
# Identifying precursors and aqueous organic aerosol formation pathways during the SOAS campaign

Neha Sareen[1], Annmarie G. Carlton[1], Jason D. Surratt[2], Avram Gold[2], Ben Lee[3], Felipe D. Lopez-Hilfiker[3, a], Claudia Mohr[3, b], Joel A. Thornton[3], Zhenfa Zhang[2], Yong B. Lim[1, c], Barbara J. Turpin[2*]

1. Department of Environmental Sciences, Rutgers University, 14 College Farm Road, New Brunswick, New Jersey 08901, United States

2. Department of Environmental Sciences and Engineering, Gillings School of Public Health, University of North Carolina at Chapel Hill, Chapel Hill, North Carolina 27599, United States

3. Department of Atmospheric Sciences, University of Washington, Seattle, Washington 98195 United States

[a] Now at Laboratory of Atmospheric Chemistry, Paul Scherrer Institute, 5232 Villigen PSI, Switzerland

[b] Now at Institute of Meteorology and Climate Research, Atmospheric Aerosol Research, Karlsruhe Institute of Technology, Karlsruhe, Germany

[c] Now at Center for Environment, Health and Welfare Research, Korea Institute of Science and Technology, Seoul 02792, Republic of Korea

*bjturpin@email.unc.edu

*Correspondence to*: Neha Sareen (neha.sareen15@gmail.com)

**Abstract.** Aqueous multiphase chemistry in the atmosphere can lead to rapid transformation of organic compounds, forming highly oxidized low-volatility organic aerosol and, in some cases, light-absorbing (brown) carbon. Because liquid water is globally abundant, this chemistry could substantially impact climate, air quality, and health. Gas-phase precursors released from biogenic and anthropogenic sources are oxidized and fragmented, forming water-soluble gases that can undergo reactions in the aqueous phase (in clouds, fogs, and wet aerosols) leading to the formation of secondary organic aerosol ($SOA_{AQ}$). Recent studies have highlighted the role of certain precursors





like glyoxal, methylglyoxal, glycolaldehyde, acetic acid, acetone, and epoxides in the formation of $SOA_{AQ}$. The goal of this work is to identify additional precursors and products that may be atmospherically important. In this study, ambient mixtures of water-soluble gases were scrubbed from the atmosphere into water at Brent, Alabama during the 2013 Southern Oxidant and Aerosol Study (SOAS). Hydroxyl (OH•) radical oxidation experiments were conducted with the aqueous mixtures collected from SOAS

to better understand the formation of SOA through gas-phase followed by aqueous-phase chemistry. Total aqueous-phase organic carbon concentrations for these mixtures ranged from 92-179 µM-C, relevant for cloud and fog waters. Aqueous OH-reactive compounds were primarily observed as odd ions in the positive ion mode by electrospray ionization mass spectrometry (ESI-MS), indicative of alcohols, carbonyl compounds, and/or

epoxides. Ultra high-resolution Fourier-transform ion cyclotron resonance mass spectrometry (FT-ICR-MS) spectra and tandem MS (MS/MS) fragmentation of these ions were consistent with the presence of carbonyls and tetrols. Products were observed in the negative ion mode and included pyruvate and oxalate, which were confirmed by ion chromatography. Pyruvate and oxalate have been found in the particle phase in many

locations (e.g., as salts and complexes). Thus, formation of pyruvate/oxalate suggests the potential for aqueous processing of these ambient mixtures to form $SOA_{AQ}$.

## 1 Introduction

     Aqueous multiphase chemistry has the potential to alter the climate-relevant

properties and behavior of atmospheric aerosols. It is well established that a major pathway for secondary organic aerosol (SOA) formation is via the partitioning of semi-volatile products of gas-phase photochemical reactions into preexisting organic particulate matter (Seinfeld and Pankow, 2003). Semi-volatile partitioning theory is widely used to model SOA (Odum et al., 1996; Seinfeld and Pankow, 2003; Donahue et

al., 2006). However differences between organic aerosol mass/properties predicted via



this formation mechanism and measured in the atmosphere suggest that other processes (e.g., aqueous chemistry) may also contribute (Foley et al., 2010; Hallquist et al., 2009).

Recent studies have highlighted the importance of water-soluble organic gases (WSOGs), liquid water, and condensed-phase reactions to SOA formation and properties (Ervens et al., 2011; Monge et al., 2012; Carlton and Turpin, 2013). Biogenic and anthropogenic gas-phase precursors are oxidized to form WSOGs such as glyoxal, methylglyoxal, glycolaldehyde, and acetone (Spaulding et al., 2003). These WSOGs are too volatile to form SOA through absorptive partitioning, but they can undergo aqueous reactions in clouds, fogs, and wet aerosols to form low-volatility products and "aqueous" SOA ($SOA_{AQ}$) (Blando and Turpin, 2000; Ervens et al., 2004; Kroll et al., 2005; Liggio et al., 2005; Lim et al., 2005; Heald et al., 2006; Loeffler et al., 2006; Sorooshian et al., 2006; Volkamer et al., 2006; Volkamer et al., 2007; Ervens et al., 2008; De Haan et al., 2009a; El Haddad et al., 2009; Ervens et al., 2011; Rossignol et al., 2014). Inclusion of aqueous chemistry of clouds, fogs, and wet aerosols in models and experiments helps to explain SOA discrepancies, particularly high atmospheric O/C ratios, enrichment of organic aerosol aloft, and formation of oxalate, sulfur- and nitrogen-containing organics and high molecular weight compounds (Kawamura and Ikushima, 1993; Kawamura et al., 1996; Crahan et al., 2004; Kalberer et al., 2004; Herrmann et al., 2005; Altieri et al., 2006; Carlton et al., 2006; Heald et al., 2006; Volkamer et al., 2007; Nozière and Cordova, 2008; De Haan et al., 2009b; El Haddad et al., 2009; Galloway et al., 2009; Shapiro et al., 2009; Volkamer et al., 2009; Lim et al., 2010; Lin et al., 2010; Nozière et al., 2010; Perri et al., 2010; Sareen et al., 2010; Schwier et al., 2010; Sorooshian et al., 2010; Sun et al., 2010; Ervens et al., 2011; Lee et al., 2011; Tan et al., 2012; Ervens et





al., 2013; He et al., 2013; Gaston et al., 2014; Ortiz-Montalvo et al., 2014). Although

uncertainties are large, modeling studies show that $SOA_{AQ}$ is comparable in magnitude to

"traditional" SOA (Carlton et al., 2008; Fu et al., 2008; Fu et al., 2009; Gong et al., 2011;

Myriokefalitakis et al., 2011; Liu et al., 2012; Lin et al., 2014). However, $SOA_{AQ}$

precursors and their chemical evolution remain poorly understood.

Much of what we know about aqueous chemistry leading to $SOA_{AQ}$ formation is

derived from laboratory studies with single precursors hypothesized to be important;

however, the most important precursors for $SOA_{AQ}$ formation in the ambient

environment may remain unidentified. A small number of studies conducted with

ambient mixtures have provided insights into the pathways of $SOA_{AQ}$ formation. For

example, photochemical oxidation of aerosol filter samples and cloud water from

Whistler, British Columbia suggest that water-soluble organic compounds of intermediate

volatility (e.g. *cis*-pinonic acid) can be important precursors for $SOA_{AQ}$ (Lee et al., 2012).

Pyruvic acid oxidation experiments in Mt. Tai, China cloud water suggested a slowing of

pyruvic acid oxidation presumably due to competition for OH radicals with the complex

dissolved cloud water organics (Boris et al., 2014). However, further work is needed to

identify precursors important for ambient $SOA_{AQ}$ formation in atmospheric waters.

This work reports, for the first time, results of aqueous OH radical oxidation

experiments conducted in ambient mixtures of water-soluble gases. Ambient mixtures

were collected in the Southeast US during the Southern Oxidant and Aerosol Study

(SOAS) in the summer of 2013; experiments were used to identify water-soluble gases

that may serve as precursors of atmospheric aqueous SOA. This region has experienced

an overall cooling trend in surface temperature over the second half of the twentieth





century, compared to the warming trend observed elsewhere in the US (Robinson et al., 2002; Goldstein et al., 2009; Portmann et al., 2009). Biogenic sources dominate emissions in this region with varying degrees of impact from anthropogenic sources.

Measurements by Nguyen et al. (2014) and model results by Carlton and Turpin (2013) indicate the significance of (anthropogenic) aerosol liquid water (ALW) in this region and support a role for ALW in $SOA_{AQ}$ formation. In the Southeast US, photochemistry and abundant liquid water coexist, making it an ideal location to study $SOA_{AQ}$ formation through gas-phase followed by aqueous-phase chemistry. The objective of this work is to

identify WSOGs important to $SOA_{AQ}$ formation. Since OH oxidation experiments were conducted in dilute solution, we will also identify products expected through cloud/fog processing of ambient WSOG mixtures. Products may differ in aerosols, where solute concentrations are higher, and radical-radical chemistry and acid-catalyzed reactions (e.g. epoxide ring-opening reactions yielding tetrols and organosulfates from IEPOX) are

important. We expect that aqueous chemistry in clouds, fogs and wet aerosols is a sink for reactants identified herein, and that this work will motivate laboratory studies and chemical modeling of newly identified aerosol/cloud precursors.

## 2 Methods

120   Samples collected in mist chambers during the SOAS field study were used to conduct controlled aqueous OH radical oxidation experiments.

### 2.1 Mist chamber field sampling at SOAS (in Brent, AL)





Water-soluble gases were scrubbed from filtered ambient air at the Centerville

ground site in Brent, AL during SOAS. Samples were collected from June 1 – July 14,

2013. Four mist chambers (Anderson et al., 2008a; Anderson et al., 2008b; Dibb et al.,

1994; Hennigan et al., 2009) were operated in an air-conditioned trailer (indoor

temperature, 25°C) at 25 L min$^{-1}$ in parallel for 4 hours, typically 2-3 times each day

between 7 AM and 7 PM CDT. Particles were removed by passing the ambient air

through a pre-baked quartz fiber filter (QFF) (Pall, 47mm) prior to introduction into the

mist chamber. The QFF will remove particles. In the early stages of sampling, on the

clean filter, adsorption of gases on the filter will reduce the concentrations of gases

sampled by the mist chamber until these gases reach gas phase – adsorbed phase

equilibrium. Using glyoxal as a WSOG-surrogate and the work of (Mader and Pankow,

2001) we predict that the measured WSOG in the mist chamber will be depleted for less

than 2% of our sampling time (after <0.1 m$^3$). Thus, we expect water-soluble gases to

penetrate through the QFF very efficiently for collection in the mist chamber water.

The mist chambers were operated with 25 mL of 17.5 ± 0.5 MΩ ultra-pure water;

additional water was added during the run to replace water lost by evaporation. Samples

from all four mist chambers were composited daily and frozen in 35 – 40 ml (experiment-

sized) aliquots. Total organic carbon (TOC) concentrations ranged from 45 – 180 μM-C

(Supplementary Table S1). Prior to and at the end of a sampling day, each mist chamber

was cleaned using a 5-minute DI water wash step. Additionally, after the rinse step, a 2-

minute dynamic blank run was conducted with activated carbon adsorbent upstream of

the mist chamber to remove organic gases in the flow stream.



Based on daily forecast predictions, certain days were selected for intensive sampling (Supplementary Table S1). Intensive sampling during SOAS was conducted on days when high levels of isoprene, sulfate, and $NO_x$ were predicted by the National Center for Atmospheric Research (NCAR) using the Flexible Particle dispersion model

(FLEXPART) (Stohl et al., 2005) and the Model for Ozone and Related Chemical Tracers (MOZART) (Emmons et al., 2010). In general, mist chamber samples on intensive sampling days had higher organic content (TOC = 92-179 μM-C), and hence we focused our experiments on those days (Table 1).

Two pieces of evidence suggest that gas-aqueous partitioning of the water-soluble

organic gases is close to Henry's Law equilibrium in our samples. In previous testing conducted in a different East Coast location that is a recipient of long-range transport (central New Jersey) we found that WSOG concentrations in the mist chamber leveled off after 1-3 hours of ambient sampling, suggesting that the collected WSOG mixture approaches Henry's law equilibrium over these collection times. This is consistent with

the one measurement we have of breakthrough at SOAS, where we ran two mist chambers in series and found TOC concentrations within +/- 11% of each other. Clearly quantification of individual species will require further work. However, these measurements suggest that the mist chamber samples contain a representative mixture of water-soluble gases.


## 2.2 Aqueous OH radical oxidation in a cuvette chamber

Ambient SOAS field samples were exposed to OH radicals in a custom built photochemical temperature-controlled (25°C) quartz cuvette reaction chamber. Ten



screw-capped quartz cuvettes (Spectrocell Inc., Oreland, PA) containing 3 mL of sample

were placed equidistant around a 254 nm mercury lamp (Heraeus Noblelight, Inc. Duluth,

GA) housed in a quartz sheath (Ace Glass Inc., Vineland, NJ). A solar spectrum lamp

was not used because the objective was to produce OH radicals by $H_2O_2$ photolysis,

rather than to mimic tropospheric photolysis. The chamber was protected from ambient

light by covering in aluminum foil. OH radicals ($1.25 \times 10^{-2}$ μM [OH] $s^{-1}$) were generated

*in situ* by photolysis of 125 μM $H_2O_2$, added to each cuvette prior to inserting the lamp.

Cuvettes were removed at t = 10, 20, 30, 40, 60, 80, 100, 120, 150 min and any

remaining $H_2O_2$ was destroyed by addition of 36 μL of 1% catalase (Sigma; 40,200

units/mg). A duplicate cuvette was removed at t=40 min to calculate method precision.

The following control experiments were performed: 1) sample + $H_2O_2$, 2) sample + UV,

3) $H_2O_2$ + UV, and 4) field water blank + OH. Replicate experiments were performed on

selected samples. Ambient conditions for sample collection are given in Table 1 for

samples used in experiments.

### 2.3 Analytical methods

185       Samples and field water blanks from all collection days were characterized by

total organic carbon analysis (TOC; Shimadzu 5000A) and electrospray ionization mass

spectrometry (ESI-MS; HP – Agilent 1100). Ion Chromatography (IC; Dionex ICS 3000)

was used to analyze organic anions and track the formation of products and

intermediates. Samples at each reaction time were analyzed by ESI-MS in positive and

negative ion modes to identify precursors and products. Selected samples were also

analyzed by ultra-high resolution electrospray ionization Fourier-transform ion cyclotron



resonance mass spectrometry (ESI-FT-ICR-MS) and tandem MS (MS-MS) on a Thermo-Finnigan LTQ-XL at Woods Hole Oceanographic Institute, MA to determine elemental composition and extract structural information on precursors. Analytical details and

quality control measures have been described previously (Perri et al., 2009).

Briefly, the ESI quadrupole mass spectrometer was operated in positive and negative ion modes over a mass range of 50-1000 amu. In the negative ion mode, the mobile phase consisted of 1:1 methanol/0.05% formic acid in water; and in the positive ion mode, 0.05% formic acid in water. The fragmentor and capillary voltages for the ESI-

MS were set at 40 V and 3000 V (nitrogen drying gas; 10 L min$^{-1}$; 350 °C), respectively. Nitrobenzoic acid in the negative ion mode and caffeine in the positive ion mode were used as mass calibrants. Standard mixtures were analyzed with each experimental sequence: acetic acid, pyruvic acid, nitric acid, succinic acid, tartaric acid, ammonium sulfate, and oxalic acid in the negative ion mode and glyoxal, methylglyoxal, and

glycolaldehyde in the positive ion mode.

ESI ionization efficiency varies with sample mix and over time. However, the mass spectra of the mist chamber samples were similar across experimental days and the variability in the glyoxal standard ESI signal was <6% across analysis days, suggesting that ion abundance trends will reflect concentration trends. Six injections were averaged

for each sample and data retained for each ion abundance greater than zero within 95% confidence intervals. Ions were considered to be above detection limits if greater than the average plus three standard deviations of the water blank.

Organic acids were measured by IC (IonPac AS11-HC column; 30 °C, AG11-HC guard column) with conductivity detection (35 °C), using a milli-q water eluent and KOH



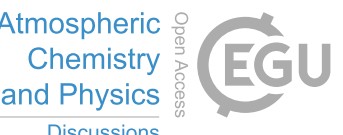

gradient method. For oxalate, the method precision is 22%, calculated as a pooled

coefficient of variation (CV) from pairs of cuvettes removed at t=40 min.  The analytical

precision for oxalate is 19% (pooled CV) based on replicate analysis of 30% of samples.

Analytical accuracy for oxalate is 7%. The limit of detection  (LOD) for oxalate by this

protocol has been previously determined to be 0.1 µM (Perri et al., 2009).

220          Samples from June 15 and June 30, 2013 were analyzed using ultra-high

resolution FT-ICR-MS in the positive ion mode using 1:1 methanol/water as the mobile

phase at 4 µL min$^{-1}$, capillary temperature of 260 °C and spray voltage 3.8-4.2 kV.

Weekly analysis of standards (caffeine, peptide-MRFA, ultramark, SDS, and sodium

taurocholate) verified the mass accuracy < 2 ppm. Note, mass accuracy tends to decrease

with decreasing $m/z$, so the accuracy of masses smaller than our smallest standard (195

amu) are likely to be lower. Previously pyruvic acid and peroxyhemiacetal standards

analyzed with the same protocol were within 2 - 10 ppm.  Error for masses determined by

difference between parent and fragment ions is larger. Five precursor masses were

isolated (isolation width: $m/z = 2$) and fragmented by collision-induced dissociation

(CID) (normalized collision energy: 26-33%) with helium in the ion trap (IT) and

infrared multi photon dissociation (IRMPD) with a $CO_2$ laser. Midas Molecular Formula

Calculator (v1.1) was then used on the mass of each detected peak to determine elemental

composition (within ±1 ppm) and calculate double bond equivalence (DBE). No

restrictions were placed on the number of carbon, hydrogen, oxygen, nitrogen, sodium

and sulfur included in the molecular formula calculations.

          Compounds are detected in the ESI-MS by forming cluster ions with hydrogen,

sodium, or ammonium in the positive ion mode; compounds are sometimes hydrated with

water or methanol. In the negative ion mode, ions are deprotonated. The presence of an even or odd mass provides insight into the elemental formula of the compound. For

instance, an even mass is an indication of a nitrogen-containing compound due to the addition of hydrogen to the odd numbered bonding of nitrogen (e.g. N + 3H + 1H$_{ionization}$ =18 amu).

## 3 Results & Discussion

OH oxidation experiments at concentrations relevant to cloud/fog water were conducted on samples collected June 11, 12, 15, 16, 20, 21, 29, and 30 of 2013, days on which total organic carbon (TOC) was highest, ranging from 92-179 µM-C in samples.

### 3.1 Precursors in SOAS samples


The concentration dynamics in experiments conducted with the 8 daily composites were similar. Positive ions at *m/z* 125, 129, 143, 173, and 187 exhibited reactant-like trends (Figure 1; June 30 sample + OH), showing decreasing signal intensity with longer exposure to OH. All the peaks disappeared after 40 minutes of oxidation. In

control experiments the abundance of these ions did not change over time, as illustrated in Figure 1 (sample + UV and sample + H$_2$O$_2$) for the positive ion at *m/z* 187 in samples collected on June 15 and 30. Hence, reaction with UV or H$_2$O$_2$ alone does not explain the decreasing signals in the presence of OH radical. Experiments conducted on all sample days showed the same reactants decreasing with exposure to OH, indicating that the



water-soluble organics captured from the ambient daytime air in the mist chambers varied

little across the study.

As discussed above, all precursor ions appear in the positive ion mode of the ESI-

MS, consistent with the presence of carbonyl compounds and polyols . They are odd ions,

suggesting they are likely not nitrogen-containing species. Elemental formulae assigned

to precursor ions (using Midas) and MS-MS fragmentation data for the corresponding

ions from June 15 and 30 samples are shown in Table 2. Both sampling days showed

similar fragmentation spectra, consistent with the presence of the same compounds on

both days, despite potential differences in the air mass from these days. No fragmentation

occurred for positive ions at *m/z* 143, 129 and 125. Proposed structures for the positive

ions at *m/z* 187 and 173 are shown in Figures 2 and 3. Possible structures for the other

ions are shown in Figure 5. As seen in the proposed structures, the parent compounds

contain multiple –OH groups, suggesting that they are polyols or aldehydes in the gas-

phase and are hydrated with water or methanol in our ESI-MS. The O:C ratio for the

reactant masses range from 0.1 - 0.8. In the following sections, we discuss individual

reactants observed in the ESI-MS.

*m/z 187*: This is the ion with the largest *m/z* we observe with the ESI-MS in the

positive ion mode that follows a precursor-like trend (Figures 1 and 2). Using the exact

mass of 187.0942 from the FT-ICR MS, the Midas molecular formula calculator gives a

formula of $C_7H_{16}O_4Na$, corresponding to the composition $C_7H_{16}O_4$ for the neutral

compound. (The discussion below is focused on the molecular formulas, rather than the

detected ions, i.e. $Na^+$ or $H^+$.) The daughter ion peak seen during fragmentation of the

positive ion at *m/z* 187 corresponds to *m/z* 155.0680, $CH_3OH$ loss, corresponding to the





molecular formula, $C_6H_{12}O_3$. We propose that this compound is present in the gas-phase as the $C_6H_{12}O_3$ aldehyde and in water as the $C_6H_{14}O_4$ tetrol shown in Figure 2. $C_6H_{12}O_3$ is

consistent with an oxidation product of $E$-2-hexenal and $Z$-3-hexenal, both being unsaturated aldehydes that have frequently been detected during field studies and are emitted to the atmosphere from vegetation due to leaf wounding (O'Connor et al., 2006). The oxidation mechanism of these two green leaf volatiles to form $C_6H_{12}O_3$ is shown in Figure 3.

***m/z 173*:** On most sampling days this reactant mass has the highest ion abundance in the ESI-MS operated in positive ion mode (Supplementary Figure S1). Similar to the other peaks, it reacts away within the first 40 minutes of exposure to OH in the cuvette chamber (Figure 1). The Midas-suggested molecular formula for this parent ion mass (*m/z* 173.0782) and its two fragment ions at *m/z* 141.0523 and 129.0524 are $C_6H_{14}O_4$,

$C_5H_{10}O_3$, and $C_4H_{10}O_3$, respectively (a reactive parent ion with the formula $C_4H_{10}O_3$ was also observed, and is discussed below).

A compound with a molecular formula of $C_5H_{10}O_3$ (the same mass as a fragment ion discussed above) was observed in the gas-phase in the same location (Centerville field site) by high-resolution time-of-flight chemical ionization mass spectrometry

(HRToF-CIMS) (Lee et al., 2014), coupled to a filter inlet for gases and aerosols (FIGAERO) (Lopez-Hilfiker et al., 2015; Lopez-Hilfiker et al., 2014). This could be the same compound that we measure as *m/z* 173.0782, as explained below. Note that the HRToF-CIMS employed iodide ionization, which forms organic-iodide adducts, resulting in a virtually fragmentation free ionization. Gas phase measurements from the HRToF-

CIMS were made in real time through a 3/4" PTFE inlet operated at 16 standard L min[-1].



Isoprene hydroxy hydroperoxide (ISOPOOH) and isoprene epoxide (IEPOX) are both detected at this mass, but HRToF-CIMS is more sensitive to ISOPOOH. ISOPOOH is an OH oxidation product of isoprene, which is further oxidized by OH under low-NO conditions to form isomeric isoprene epoxydiols (IEPOX) (Paulot et al., 2009). Both IEPOX and ISOPOOH are prevalent at the SOAS ground site due to the abundance of isoprene emissions in this forested region, but it is likely that $m/z$ 173 is not indicative of these two compounds in our samples. As discussed below in detail, based on ESI-MS measurements with an IEPOX standard, it can be confirmed that IEPOX was not detected in the mist chamber samples. The O-O peroxide bond in ISOPOOH is the weakest bond in the molecule, and hence when undergoing MS-MS, should be the first to fragment. There is no evidence of this bond breaking in the fragmentation spectra for $m/z$ 173, leading to the conclusion that the detected compound is not ISOPOOH. IEPOX and ISOPOOH were present in the ambient air at the field site. We expect that they were lost during sampling or storage.

A likely structure for positive mode $m/z$ 173 is shown in Figure 4. In this case the compound is proposed to be a $C_5H_{10}O_3$ aldehyde in the gas phase and a $C_5H_{12}O_4$ tetrol in water. In the FT-ICR-MS it is seen hydrated with methanol. The parent ion at $m/z$ 173 loses methanol to form $C_5H_{10}O_3$ ($m/z$ 141.0523), and it also loses $C_2H_4O$ to form $C_4H_{10}O_3$ ($m/z$ 129.0524). $C_5H_{10}O_3$ is consistent with the oxidation product of $(E)$-2-methyl-2-butenal, another green leaf volatile (Figure 3c) (Jiménez et al., 2009; Lanza et al., 2008). It has also been reported as an isoprene oxidation product (Yu et al., 1995).

**_Positive ions at m/z 143, 129, and 125_**: No fragments were observed for _these_ reactants with MS-MS. The Midas-predicted molecular formulae for the ions at $m/z$



143.0676, 129.0520, and 125.096 are $C_5H_{12}O_3$, $C_4H_{10}O_3$, and $C_8H_{12}O$, respectively.

$C_5H_{12}O_3$ and $C_4H_{10}O_3$ were also seen in the gas phase at the Centerville field site by

HRToF-CIMS. Possible structures for these compounds based on their elemental formula

and double bond equivalents are shown in Figure 5. Interestingly, the reactant detected at

$m/z$ 129 has the same mass as a fragment of the parent ion at $m/z$ 173 discussed earlier

(and the structure of the $C_4H_{10}O_3$ fragment shown in Figure 4 is another possible structure

for $m/z$ 129).

Figure S2 shows time series of ion abundance in the aqueous mist chamber

samples (8-12 hr integrated samples) and gas-phase signals of compounds with

corresponding molecular formulas as measured by HRToF-CIMS (Lee et al., 2014;

Lopez-Hilfiker et al., 2014). There are not strong correlations, but there are similarities in

trends. Note that ionization of compounds in the ESI-MS and CIMS varies with

compound class and with the composition of the mixture (matrix); instrument sensitivity

varies daily. In some cases, more than one isomer (with the same mass) may contribute to

the observed signal and isomers with the same mass may contribute differently to the

CIMS and ESI-MS signal strength (e.g., $m/z$ 173). Also, the ESI-MS measurements were

made after collection in water, whereas CIMS measurements were made in the gas phase.

Thus, there are limitations to quantitative comparisons between these measurements in

the absence of authentic standards. However, measurement of the same masses and the

similarities in trends suggest that we may be measuring the same or similar species.

We expected IEPOX and glyoxal to be present in the samples since they were

detected in the gas-phase during the SOAS campaign (Nguyen et al., 2015) (glyoxal by

Keutsch group) and have relatively high Henry's Law constants, but we did not detect



either of these compounds in our samples using our ESI-MS. Average gas-phase concentrations of IEPOX and glyoxal were measured to be ~0.09 ppb and 0.1 ppb, respectively (Nguyen et al., 2015) (glyoxal by Keutsch group), which based on their

Henry's Law constants ($H_{L,IEPOX}$=2.7 x $10^6$ M atm$^{-1}$, $H_{L,GLYOXAL}$=3.6 x $10^5$ M atm$^{-1}$) correspond to ~250 μM IEPOX and 36 μM glyoxal in the aqueous phase. When the ambient samples were spiked with 3000 μM, 300 μM, and 30 μM standards of either compound, they were readily detected, indicating that they can be ionized in our sample matrix. (Authentic *trans*-β-IEPOX, which is the predominant isomer of IEPOX, was

synthesized for this purpose (Zhang et al. (2012).) It is quite likely that these compounds are too unstable to persist through collection and storage in our aqueous samples. In fact, they may have oxidized during collection, since water-soluble oxidants would have also been scrubbed by the mist chambers).

**3.2 Product formation during aqueous oxidation experiments**

Figure 6 shows significant formation of oxalate and pyruvate in OH radical experiments conducted with all samples but not during the control experiments (Sample + UV; Sample + $H_2O_2$). Pyruvate peaks around 60-80 min, which is earlier than the oxalate

peak at 100-120 min (Figure 6). Acetate + glycolate (which co-elute in the IC) also forms in at least some samples and reacts away in the presence of OH (Supplementary Figure S3). Sulfate and nitrate concentrations remained constant throughout the experiment as measured by the IC. While there may be many sources of oxalate, aqueous OH radical oxidation of pyruvate in the aqueous phase is known to form oxalate at dilute (cloud-




relevant) concentrations (Carlton et al., 2006). Aqueous acetate oxidation is also a source

of oxalate (Tan et al., 2012). The concentration dynamics are consistent with a role for

these compounds in the formation of oxalate in the ambient mixtures although the

mechanisms by which pyruvate and acetate formed are not well constrained in these

experiments. These observations suggest that oxalate, pyruvate, and acetate can form in

ambient mixtures of water-soluble gases in the Southeast US in the presence of

clouds/fogs and oxidants. Pyruvate and oxalate have been observed primarily in the

particle phase in the atmosphere (Saxena and Hildemann, 1996; Limbeck et al., 2001;

Yao et al., 2002; Kawamura et al., 2003). Moreover, modeling studies of oxalate, the

most abundant dicarboxylic acid in the atmosphere, suggest that aqueous chemistry is a

large contributor of oxalate formation globally, making it a good tracer for $SOA_{AQ}$

formed in clouds and fogs (Myriokefalitakis et al., 2011). Above versus below cloud

measurements also support this (Sorooshian et al., 2010). Thus, the experiments suggest

that aqueous oxidation of ambient (Southeastern US) water-soluble mixtures at cloud/fog

relevant concentrations has the potential to form material that remains in the particle-

phase species after droplet evaporation, i.e. $SOA_{AQ}$. However, the atmospheric

prevalence of *particle-phase* oxalate can only be explained by the formation of salts and

complexes, since oxalic acid is volatile and the volatility of oxalate salts is orders of

magnitude lower than that of oxalic acid (Ortiz-Montalvo et al., 2014; Paciga et al.,

2014). The aerosol at the SOAS ground site was acidic (campaign average pH~0.94)

(Guo et al., 2015) and as a consequence oxalic acid may remain largely in the gas phase

in this environment, but may eventually react on more basic surfaces, e.g., coarse

particles. Note that we expect the products of aqueous chemistry in wet aerosols to be

different from those in clouds and fogs because of the extremely high (molar) solute

concentrations in wet aerosols (Surratt et al., 2007; Noziere et al., 2008; Galloway et al.,

2009; Lim et al., 2010; Sareen et al., 2010; Nguyen et al., 2012).

### 3.3 Atmospheric implications

We have tentatively characterized several water-soluble OH-reactive species

collected at an isoprene-rich photochemically-active location in the southeastern U.S. In

several cases compounds with the same elemental composition were measured in the gas

phase by HRToF-CIMS. The tentative structures for the proposed reactants are consistent

with formation from green leaf volatiles and isoprene oxidation. Aqueous OH oxidation

under dilute conditions (TOC approx. 100 μM) relevant to fogs and clouds produced

oxalate and pyruvate suggesting that cloud/fog processing of these compounds (and

subsequent neutralization or complexation) is a potential source of SOA. The reactants

characterized in this work are precursors for aqueous chemistry and are potentially

important SOA$_{AQ}$ precursors in all atmospheric waters, i.e. clouds, fogs, and wet aerosols.

The aqueous chemistry of these precursors is poorly understood and warrants further

study.


### 4 Conclusions

We observed formation of pyruvate, oxalate, and acetate/glyocolate during OH

oxidation experiments conducted with ambient mixtures of WSOG from the southeastern

US. The formation of these highly oxygenated organic acids indicates a potential for

SOA$_{AQ}$ formation (e.g., upon neutralization with NH$_3$, metal complexation or

heterogeneous reaction on course dust/salt particles). Given the acidity of SOAS fine particles, we think it is unlikely that oxalate will be found in substantial quantities in the fine aerosol at the SOAS ground site.

We tentatively characterized several water-soluble reactive precursors of aqueous chemistry and $SOA_{AQ}$ formation in wet aerosols, clouds and fogs at this location. High resolution mass spectrometric analyses suggest precursors had O:C ranging from 0.125-0.80 and several of them could plausibly be tied to isoprene oxidation. No distinct difference was seen in the aqueous oxidation of ambient samples collected across days during the SOAS field campaign. Further work involving organic synthesis, aqueous OH oxidation of authentic standards, and mass spectral analyses with pre-separation are likely to yield further insights into the aqueous chemistry of these compounds in the future.

## 5 Acknowledgements

This research was funded by the EPA STAR grant 835412. The views expressed in this manuscript are those of the authors and do not necessarily reflect the views or policies of the U.S. Environmental Protection Agency. The authors acknowledge Melissa Soule and funding sources of the WHOI FT-MS User's Facility (NSF OCE-0619608 and the Gordon and Betty Moore Foundation). We also thank Jeffrey Kirland, Ronald Lauck, and Nancy Sazo for their invaluable assistance in the laboratory and field, and Louisa Emmons for the air quality forecasting during SOAS.

Supporting Information Available. Included are Table S1 and Figures S1-S3. The material is available free of charge via the Internet at http://pubs.acs.org.





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



**Tables & Figures**

**Table 1**: Sample collection dates for which experiments were run and their sampling

conditions. Temperature, relative humidity, and ozone ranges are shown both for the

entire day and collection time period. Total organic carbon (TOC) accuracy and precision

are verified with potassium hydrogen phthalate (KHP) standards to be better than 5%

(Perri et al., 2009).

| Collection date | Collection time | μM TOC | T (°C) all day (coll. time) | RH (%) all day (coll. time) | O₃ (ppbᵥ) all day (coll. time) |
|---|---|---|---|---|---|
| 11-Jun-2013 | 7am-7pm | 139.5 | 22-32 (23-32) | 53-99 (53-98) | 9.9-38.2 (11.3-38.2) |
| 12-Jun-2013 | 7am-7pm | 179.7 | 23-33 (23-33) | 48-94 (48-94) | 13.1-41.8 (13.1-41.8) |
| 15-Jun-2013 | 7am-7pm | 117.0 | 17-31 (18-31) | 45-94 (45-90) | 11.6-53.4 (11.6-53.4) |
| 16-Jun-2013 | 7am-7pm | 108.2 | 22-32 (22-32) | 53-93 (53-84) | 2.6-41.7 (23.1-41.7) |
| 20-Jun-2013 | 8am-5pm | 131.5 | 20-30 (21-30) | 55-98 (59-94) | 4.8-52.2 (6.2-42.8) |
| 21-Jun-2013 | 10am-6pm | 104.4 | 20-30 (25-30) | 50-93 (50-78) | 16.6-45.2 (30.3-45.2) |
| 29-Jun-2013 | 7am-7pm | 92.0 | 21-31 (22-31) | 43-100 (43-100) | 16.3-53.7 (16.3-53.7) |
| 30-Jun-2013 | 7am-7pm | 98.7 | 20-30 (20-30) | 38-100 (38-100) | 1-53.5 (1-53.5) |



**Table 2**: Elemental formulas assigned to precursor ions using ESI-FT-ICR MS in the positive ionization mode and Midas Molecular Formula Calculator. MS/MS fragmentation data is also shown.

| Precursor peak using ESI-MS | Positive mode $m/z$ (using FT-ICR) | $[M+Na]^+$ or $[M+H]^+$ | Mol. Wt. | Double bond equivalence |
|---|---|---|---|---|
| 187 | 187.0942 | $C_7H_{16}O_4Na$ | 164.1043 | 0 |
| | 155.0680 | $C_6H_{12}O_3Na$ | 132.0786 | 1 |
| 173 | 173.0782 | $C_6H_{14}O_4Na$ | 150.0887 | 0 |
| | 141.0523 | $C_5H_{10}O_3Na$ | 118.0625 | 1 |
| | 129.0524 | $C_4H_{10}O_3Na$ | 106.0625 | 0 |
| 143 | 143.0676 | $C_5H_{12}O_3Na$ | 120.0781 | 0 |
| 129 | 129.0520 | $C_4H_{10}O_3Na$ | 106.0625 | 0 |
| 125 | 125.096 | $C_8H_{13}O$ | 124.0883 | 3 |





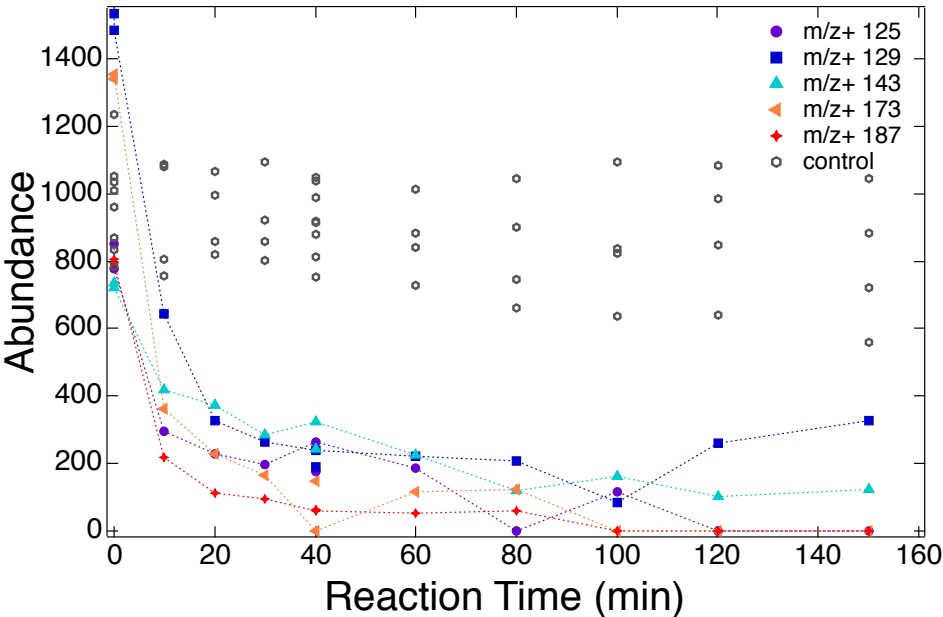


**Figure 1.** Positive ions (ESI-MS) exhibiting precursor-like trends during aqueous OH-radical oxidation experiments with the ambient mixtures collected on June 30. All days show similar trends, with all 5 reactant masses showing statistically significant decreasing trends as compared to the control experiments. Controls (sample + UV, 790    sample + $H_2O_2$) shown for *m/z* 187; other masses show similar trends.



**Figure 2**. Proposed structure for the positive ion at *m/z* 187. The top structure is the parent compound detected as a reactant in the ESI-MS; the following structures show the MS/MS fragments. The respective gas-phase and aqueous-phase compounds are shown in the shaded boxes at the bottom.



**(a)**

**(b)**

**(c)**


**Figure 3**: Gas-phase oxidation of (a) *E*-2-hexenal and (b) *Z*-3-hexenal and (c) (*E*)-2-

methyl-2-butenal.



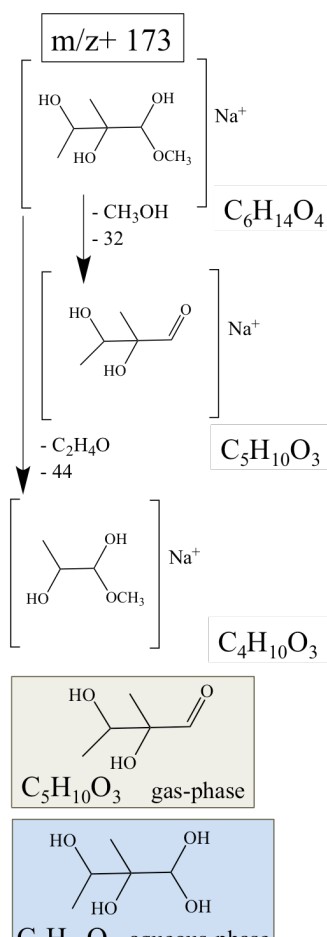


**Figure 4**. Proposed structure for the positive ion at *m/z* 173. The top structures in each panel are the parent compound detected as a reactant in the ESI-MS; the following structures show the MS/MS fragments. The respective gas-phase and aqueous-phase compounds are shown in the shaded boxes at the bottom.




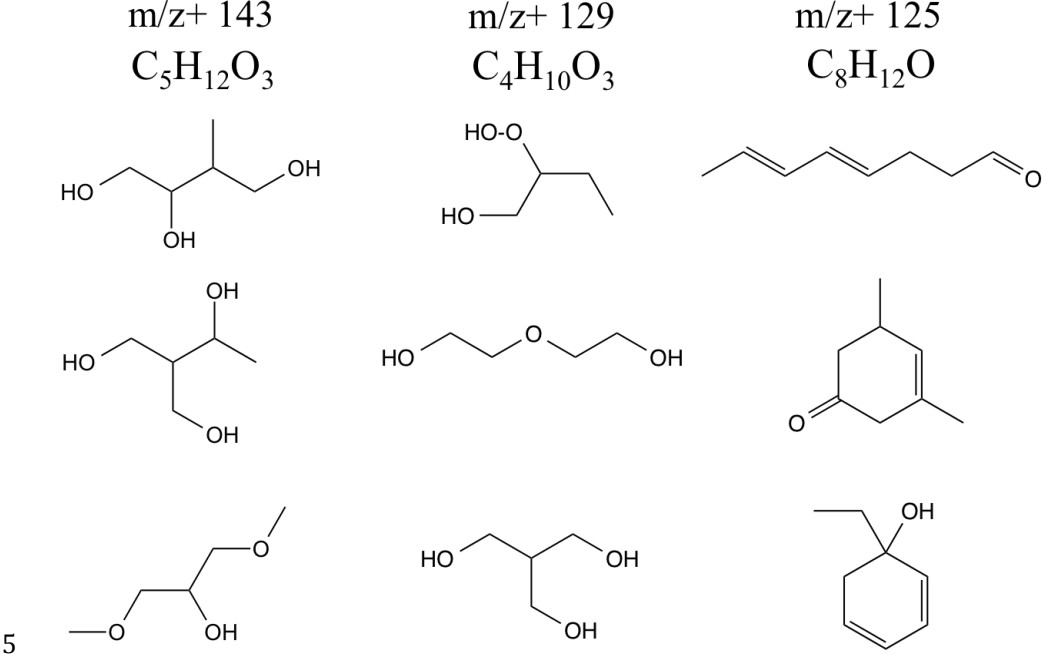


**Figure 5**: Possible structures for the positive ions at *m/z* 143, *m/z* 129, and *m/z* 125. These structures are based on MIDAS suggested elemental formulae for these masses and double bond equivalence (DBE). No fragmentation was observed for these reactant masses.





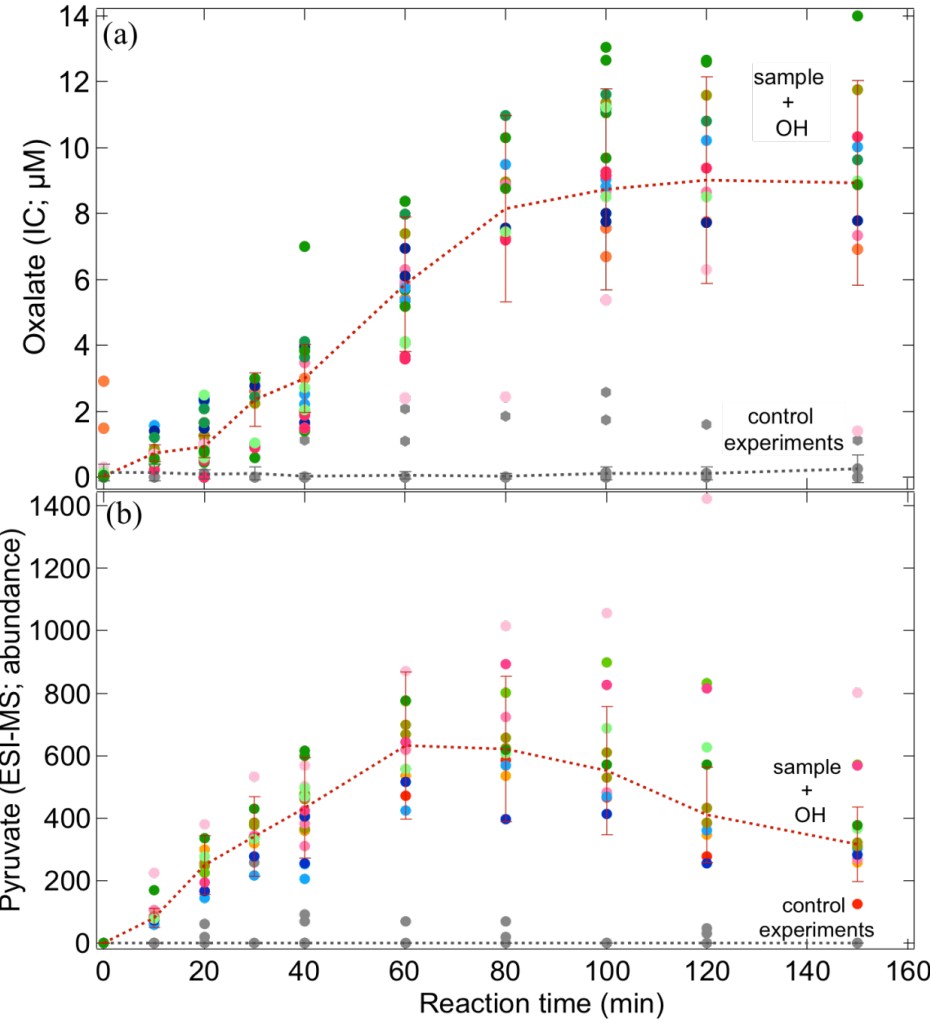


**Figure 6**. (a) Oxalate (by IC) for all OH radical oxidation experiments conducted with ambient samples (Table 1). (b) Abundance of the negative ion at *m/z* 87 (pyruvate) as observed in the ESI-MS when the ambient SOAS samples are exposed to OH. Error bars represent the pooled coefficient of variation calculated across experimental days. Note that oxalate formed in all samples in the presence, but not the absence, of OH. Gray points represent control experiments (June 11 sample + UV, June 11 sample + $H_2O_2$, June 30 field water blank + OH).