# Peer review of "Identifying precursors and aqueous organic aerosol formation pathways during the SOAS campaign"

_Atmospheric Chemistry and Physics, 2016_

## Referee Comment (RC1) · Anonymous Referee #1 · 8 May 2016

The paper describes measurements of the oxidative chemistry of atmospheric water-soluble organic compounds (WSOC). Mist chamber samples were taken during the SOAS campaign, for the collection of gas-phase WSOC. Collected organic species were then oxidized offline by OH radicals (generated by addition of $H_2O_2$ followed by UV irradiation), and oxidation chemistry was tracked using ESI-MS and IC. A small number of organic species were found to decrease with OH oxidation, and a few oxidation products were found to be formed. These results are interpreted in terms of the formation of secondary organic aerosol (SOA) within cloud or fog droplets.

This is useful study on an important topic in atmospheric chemistry. The overall approach (collection and aqueous-phase oxidation of ambient WSOC) is a novel one, and has the potential to provide insights into the role of aqueous-phase oxidation in SOA formation. However, the actual measurements described in this manuscript raise

a number of questions about the relationships of the organics collected/measured to those in the ambient atmosphere – examples include the fraction of ambient gas-phase WSOG that are actually collected, the possibility of loss of organic species during sampling, and the potential role of background organic species. These concerns are described below, and need to be addressed if this work is to be published in ACP.

1) The authors point out that there are some large discrepancies between in-situ measurements of gas-phase water-soluble organic gases (e.g., ISOPOOH, glyoxal) and the species measured within the mist chambers. This is attributed to "loss during sampling or storage" (line 319), with the possibility that they may have undergone oxidation within the water (line 360). Since such losses have major implications for the generalization of results, these need to be discussed more thoroughly. First, irreversible loss to the sample lines/inlets can be a major sink for IVOCs, so this needs to be considered. What was the sampling scheme used? (The length, diameter, and material used for the sample tubing needs to be given.) What is the sampling efficiency of water-soluble standards sent through this sampling setup? This is briefly touched on in the paper, in a discussion of glyoxal loss to the particle filter (lines 134-136), but this is based on rough calculations and not actual measurements, and doesn't consider the role of losses to tubing.

Further, if oxidation within the collected (non-irradiated) sample is occurring (lines 360-363), there is some important chemistry here that needs to be discussed. The authors appear to be arguing that glyoxal and ISOPOOH react with in-situ, non-OH oxidants ($H_2O_2$, etc). Is this consistent with previous studies, and the existing literature? Moreover such behavior would have to be different from that of compounds focused on in this study: they would have to be resistant to oxidation by any oxidants collected within the mist-chamber samples, but still reactive with OH. Is this the argument being made here? If so, what are the implications for atmospheric oxidation of these various species?

2) The paper focuses entirely with organic species that exhibit "reactant-like trends".

However there also should be some discussion of ions that do not exhibit such trends, if any. Were any of these observed? If so, what fraction of total WSOC signal do they make up? What are their formulas and characteristics? (Why are they not oxidized by OH?)

3) While there are some comparisons of mist-chamber species (collected WSOC) and CIMS (gas-phase compounds), this is only for a few select compounds. A more comprehensive comparison of the data from the two techniques is an important and necessary test of the hypothesis that these mist chambers are collecting WSOGs from the gas phase. For example, are there any WSOGs (other than ISOPOOH) measured by the CIMS (or PTRMS, etc.) that are not measured in the mist chamber? Conversely, what could explain any compounds measured in the mist chamber but not by the CIMS (e.g., m/z 125)? Finally, for the ions that are measured by both techniques, the ion intensities should be compared in all cases. (Figure S2 shows only a subset.)

4) Throughout the paper, oxidation results (sample + H2O2 + UV) are compared to results from two blanks: sample + H2O2 and sample + UV. A third one that is at least as important is oxidation (UV+H2O2) of the sample-blank (water from the mist chamber that was exposed to zero air rather than ambient air, ideally sent through the same sampling setup). This is necessary for assessing whether any organic species were introduced by the sample lines, glassware, or sample handling.

Other Comments:

Lines 68-79: This sentence would be more useful if the references were put throughout the sentence (connecting studies with individual features of SOA) rather than all at the end. This is particularly important because of the argument that there are "SOA discrepancies" involving high-MW, sulfur-containing, and nitrogen-containing species. I haven't read all 30 cited references closely, but I'm unaware of any such discrepancies (presumably between measurements and models?), since those organic types are usually not followed explicitly in models, nor are measured routinely in the ambient

atmosphere.

Line 208: Glyoxal is just one of several standard compounds added. Did all the standards show a similar behavior?

238-240, 263-264: given that an FTICR is used in this study, these discussions of the implications of unit-mass resolution m/z values seem unnecessary.

251, Fig. 1, and elsewhere: What is the approximate aqueous OH concentration in these experiments?

269-273: I think the authors mean Figure 4 instead of Figure 3 here. This whole section could be removed, since it introduces a lot of information and raises questions that are not addressed until later in the paper.

Figure 2: the positive ion (Na+) should be given here, as it is in Figure 4.

Lines 284-285, 324-325; Figure 3: This chemistry all involves the formation of diols from C-C double bonds. However diols are not generally considered to be alkene oxidation products. The main routes for formation of diols are are when RO2+RO2 reactions dominate (e.g., Ruppert and Becker 2000, Atmos. Environment, 34, 1529-1542) or in the oxidation of conjugated dienes (leading to the formation of isoprene tetrols). What is the mechanism proposed here? Similarly, have these diol species been measured in any laboratory product studies?

Line 288: These aren't "mechanisms" but simply measured compounds mapped back to potential reactants.

Lines 290-319, Figure 4: When MeOH is added to the sample (for ESI analysis), might there be an exchange between –OH and –OCH3 groups? If so, this species might simply be from an isoprene tetrol.

Figure 5: I would think some of these structures could undergo fragmentation during MS-MS analysis. As stated in line 315, O-O bonds should break in this case.

[Figure]

Figure 5: The structures given for m/z 125 are quite non-polar, with only one functional group in an eight-carbon molecule. Are they sufficiently water-soluble that they would be expected to be trapped in the mist chamber (or in cloud/fog droplets in the atmosphere)?

297-319: The argument in this paragraph was hard to follow. First it is stated that the CIMS and ESI measurements could "be the same compound". But then it's argued that CIMS is measuring IEPOX/ISOPOOH, which is not measured in the mist-chamber samples by ESI. But yet this ion is given as one of the CIMS-ESI comparisons in S2, suggesting the authors are arguing the species are in fact the same. This needs to be clarified.

Line 326: the structure given in Fig. 4 is not consistent with isoprene oxidation, since it has two methyl groups. The only plausible isoprene product (with one methyl group) of that formula that I can think of is IEPOX.

Figure S2: Plots need axis labels and units.

Lines 381-382: This observation may simply be a result of a historical lack of good techniques for measuring gas-phase pyruvate and oxalate (i.e., pyruvic acid and oxalic acid). These compounds can now be measured routinely with CIMS (with acetate ionization), so these sorts of comparisons haven't really been able to be made well until now.

Lines 426: What were the "several" isoprene products? I think only one ($C_5H_{10}O_3$) is discussed in the paper (but see my comments above).

Overall: From the concentrations of oxalate/pyruvate formation, this work should give a rough upper limit for the amount of aqueous SOA that can be formed from cloud/fog processing in a given sample (assuming collection of all gas-phase WSOC). What is this value?

---

## Referee Comment (RC2) · Anonymous Referee #2 · 23 May 2016

**1   General Comments:**

The authors of this manuscript collected gas phase compounds during the SOAS measurement campaign and measured their reaction progress in the presence of OH to understand the aqueous phase chemistry that occurs as a result of gas-to-particle partitioning of these compounds. They measured loss rates of several oxidized compounds and formation rates of four organic acids as a result of reaction of these ambient compounds with OH and monitored the formation of highly oxidized oxalate, pyruvate, acetate, and glycolate over time. This indicates that the aqueous phase oxidation of water soluble organic compounds may be very important for the formation of these compounds, possibly followed by repartitioning back to the gas phase after oxidation. This study is important for the atmospheric chemistry because it uses ambient com-
pounds as precursors for oxidation, instead of single "proxy" compounds or simple mixtures of such. The authors were able to track both precursor and product compounds, which could lead to mechanistic conclusions about the formation of organic acids in atmospheric waters. However, more details are needed about both the sampling method (e.g., recovery of compounds through the mist chambers and instrumental precision, see comments below) and about other compounds besides those discussed here. This will give the reader a better sense of the significance organic acid formation from the compounds discussed here.

The five masses that exhibited "reactant-like trends" are discussed in detail. Are these the only compounds observed in the initial mixtures? If not, were there others that reacted with OH or that stayed constant over the course of the experiment? It seems unlikely that these are the only compounds detected, and more detail would be appreciated to give the reader a sense of the concentrations of these compounds as compared to others. How significant were these five compounds in terms of the percentage of organic matter? Why wouldn't other compounds react with OH? How likely is it that compounds that are not detected contribute to organic acid formation? If these are the only compounds that displayed "reactant-like trends," then this will allow for the conclusion that oxalate, pyruvate, acetate, and glycolate are directly produced from these compounds and not from others. Otherwise, this argument is hard to make. A similar comment can be made for the oxidized species formed from the OH reaction. Only four organic acid products are mentioned in the manuscript. Were others observed? If not, is this expected? The reaction of OH with organics is likely to produce these small organic acids eventually, but I would expect to see other acids formed as intermediates in this process as well.

Discussion of the significance of these compounds in WSOC would strengthen this manuscript. As is, the authors make some interesting conclusions about the formation of aqueous SOA formation, but it is difficult to determine the magnitude of their contributions to aqueous phase chemistry. An upper bound estimate of this contribution

might also be interesting to see.

**2   Specific Comments:**

The authors state in the methods section that intensive days were chosen because of high concentration predictions by NCAR and that during those days, they observed higher than usual TOC content in general. However, in examining Supplementary Table S1, it seems that the days marked "intensive" are not much higher than others, if at all. The range given in the text (92-179 $\mu$M-C) does not match the range of TOC on intensive days. Can this be clarified? Does this range only represent the days that were used in further measurements?

Line 208: Why is glyoxal the only standard compared across all analysis days? Were the variabilities of the other standards on a similar scale? If so, this should be stated here.

Line 215-219: Limit of detection and precision information is given here for oxalate, but this is not the only organic acid measured by IC. Can all the acids studied be considered to be similar to this or do they vary significantly?

In Figure 2, *m/z* 187 is actually the sodium cluster, which is not shown in the figure. However, Figure 4 makes it clear that this is the case. These should be changed to be consistent with eachother.

A predicted structure for *m/z* 187 is shown in Figure 2, along with the predicted gas- and aqueous- phase structures. However, the observed mass contains an extra methoxy group. From previous comments on the methodology and the discussion that follows about the compound at *m/z* 173, I assume this is a result of hydration by methanol in the FT-ICR-MS. However, this is not stated until after the discussion of *m/z* 187, and should be explained the first time it is seen.

In line 319 and again in lines 360-364, the authors state that they expect IEPOX, ISOPOOH, and glyoxal were lost in their experiments during sampling or storage. Have any tests been done to quantify losses of other compounds during sampling or storage? If these compounds were lost, it seems likely that there are others that are either lost or not fully recovered. Were any recovery studies done with known amounts of standards instead of spiking the samples just before analysis?

In Figure 5, for *m/z* 125, how likely is it that the first structure shown (the aldehyde) is detected as such and not hydrated in the aqueous mixture? Is this a likely structure? The authors also state in the discussion of this figure that gas phase compounds are seen at *m/z* 129 and 143. However, it seems unlikely that these compounds are the polyols found in Figure 5. Are there any compounds with those masses that may be found in the same form in both the gas- and aqueous- phase?

In the Figure 6 caption, special mention is made of the fact that oxalate is present in all samples. Is this not true for pyruvate as well?

I'm not really sure why Supplementary Figure S3 is not included in the main portion of the manuscript. It is discussed in the main text and seems to be important to the overall conclusions of the paper. It is also hard to follow, mainly because there are so many points. It would be easier to see the trends if a general trend line was added as in Figure 6.

Acetate and glycolate are found in some samples, but in varying concentrations. Did the authors see any trends that might explain their formation on some days and not others? Also related, if acetate and glycolate co-elute on the IC, how can the statement be made that "Acetate formation is seen on some but not all days" when any signal seen is due to the combination of both? Was acetate also measured via another method? There seems to be a lack of discussion about the glycolate present. In the discussion of these results (lines 375-380), acetate is mentioned but not glycolate. Is this because it is expected that most of this combined signal is acetate or because glycolate is not

expected to be an important oxalate precursor in these ambient mixtures?

Were any measurements of oxalate in the particle phase taken at SOAS? in lines 421-423, the authors state that based on their conclusions, it is unlikely that oxalate will be present in the particle phase, but it would be interesting to test this.

**3   Technical Corrections:**

Page 6, line 140: "ml" should be "mL"

Page 7, line 159: "Henry's law" should be "Henry's Law"

Page 12, line 270: Should this refer to Figures 2 and 4 instead of 2 and 3?

Page 13, line 288: Figure 3 does not show a mechanism, but only initial and final structures.

Page 14, line 327: It is unclear to me why the word "*these*" is italicized.

Page 16, line 363: There is an extra parenthesis at the end of the paragraph.

Supplementary Figure S2: This figure is missing axes labels.

Supplementary Figure S3 caption, line 2: "co-elude" should be "co-elute"

---

## Author Comment (AC1) · 1 Aug 2016

**REVIEWER 1**

Review of "*Identifying precursors and aqueous organic aerosol formation pathways during the SOAS campaign*" by Sareen et al.

*The paper describes measurements of the oxidative chemistry of atmospheric water-soluble organic compounds (WSOC). Mist chamber samples were taken during the SOAS campaign, for the collection of gas-phase WSOC. Collected organic species were then oxidized offline by OH radicals (generated by addition of H2O2 followed by UV irradiation), and oxidation chemistry was tracked using ESI-MS and IC. A small number of organic species were found to decrease with OH oxidation, and a few oxidation products were found to be formed. These results are interpreted in terms of the formation of secondary organic aerosol (SOA) within cloud or fog droplets.*

*This is useful study on an important topic in atmospheric chemistry. The overall approach (collection and aqueous-phase oxidation of ambient WSOC) is a novel one, and has the potential to provide insights into the role of aqueous-phase oxidation in SOA formation. However, the actual measurements described in this manuscript raise a number of questions about the relationships of the organics collected/measured to those in the ambient atmosphere – examples include the fraction of ambient gas-phase WSOG that are actually collected, the possibility of loss of organic species during sampling, and the potential role of background organic species. These concerns are described below, and need to be addressed if this work is to be published in ACP.*

We thank the reviewer for their helpful feedback and suggestions. We address each point in order below.

*1) The authors point out that there are some large discrepancies between in-situ measurements of gas-phase water-soluble organic gases (e.g., ISOPOOH, glyoxal) and the species measured within the mist chambers. This is attributed to "loss during sampling or storage" (line 319), with the possibility that they may have undergone oxidation within the water (line 360). Since such losses have major implications for the generalization of results, these need to be discussed more thoroughly. First, irreversible loss to the sample lines/inlets can be a major sink for IVOCs, so this needs to be considered. What was the sampling scheme used? (The length, diameter, and material used for the sample tubing needs to be given.) What is the sampling efficiency of water-soluble standards sent through this sampling setup? This is briefly touched on in the paper, in a discussion of glyoxal loss to the particle filter (lines 134-136), but this is based on rough calculations and not actual measurements, and doesn't consider the role of losses to tubing.*
*Further, if oxidation within the collected (non-irradiated) sample is occurring (lines 360- 363), there is some important chemistry here that needs to be discussed. The authors appear to be arguing that glyoxal and ISOPOOH react with in-situ, non-OH oxidants (H2O2, etc). Is this consistent with previous studies, and the existing literature? Moreover such behavior would have to be different from that of compounds focused on in this study: they would have to be resistant to oxidation by any oxidants*

*collected within the mist-chamber samples, but still reactive with OH. Is this the argument being made here? If so, what are the implications for atmospheric oxidation of these various species?*

These suggestions to better characterize and discuss potential losses are helpful.

The length, diameter and material of the inlet tubing has been added to the first paragraph of methods section 2.1:

*"Samples were collected from June 1 – July 14, 2013 from 1 m above the trailer roof through a 1.3 cm OD PTFE inlet (approximately 1.7 m in length)."*

The reviewers' comments have been helpful in clarifying our thoughts on losses.

There are three major ways to lose water-soluble organic gases during sampling/storage:
1. Losses in tubing. Recent work by Krechmer et al highlight the uncertainties associated with quantifying the loss of gas-phase organic compounds to Teflon (Krechmer et al., 2016). This limitation is now acknowledged.
2. Ozone could also be collected during sampling and could react with certain organic gases during mist chamber collection.
3. Storage and sample handling. As described in the text, at the end of each collection day, the samples were separated into experimental sized aliquots and frozen immediately. They were shipped overnight with ice packs to the laboratory at Rutgers, where they were placed in the freezer at $-20^{\circ}C$ upon arrival. Before an experiment the sample was thawed at room temperature.

IEPOX does not survive extended storage in water (confirmed with our organic synthesis collaborator). We expect that this is also the case for ISOPOOH. This explains why we did not see IEPOX and ISOPOOH in our collected samples.

We have added a section called "Methodological Limitations" to address losses. We had already addressed losses to the quartz fiber filter, so that text has now been moved down to the new paragraph, which reads:

*"It should be noted that WSOGs can be lost during sampling and storage through: 1) losses in tubing and by adsorption to the QFF during collection, 2) reactions in the mist chamber during collection with water-soluble ambient oxidants capable of penetrating the inlet (e.g., ozone), and 3) losses during storage post collection. The QFF removes particles upstream of the mist chambers. In the early stages of sampling, on the clean filter, adsorption of gases on the filter will reduce the concentrations of gases sampled by the mist chamber until these gases reach gas phase – adsorbed phase equilibrium. Using glyoxal as a WSOG-surrogate and the work of (Mader and Pankow, 2001) we predict that the measured WSOG in the mist chamber will be depleted for less than 2% of our sampling time (after <0.1 m$^3$) due to loss to the filter. Thus, we expect water-soluble gases to penetrate through the QFF very efficiently for collection in the mist chamber water. Losses to Teflon inlets and chamber walls (Krechmer et al., 2016)*

*can be significant and variable and may reduce the number of species we are able to collect and identify in this work. While OH radicals are unlikely to penetrate the inlet, ozone might. Thus some ozone could be scrubbed by the mist chambers and could result in oxidation of some WSOGs during collection. Though many organics are stable when stored frozen in water, IEPOX does not survive extended storage in water (confirmed with our organic synthesis collaborator). We expect this to be the case for ISOPOOH also. ISOPOOH is an OH oxidation product of isoprene, which is further oxidized by OH under low-NO conditions to form isomeric IEPOX (Paulot et al., 2009). Both IEPOX and ISOPOOH are prevalent at the SOAS ground site due to the abundance of isoprene emissions in this forested region. These losses constitute the major limitation of the work in that they restrict the number of reactive water-soluble compounds we are able to identify."*

**2) The paper focuses entirely with organic species that exhibit "reactant-like trends". However there also should be some discussion of ions that do not exhibit such trends, if any. Were any of these observed? If so, what fraction of total WSOC signal do they make up? What are their formulas and characteristics? (Why are they not oxidized by OH?)**

As the reviewer suggests, we do observe other organic species (not discussed in the text) that do not react with OH. Woods Hole high-resolution instrument time is expensive and valuable. We used our instrument time to focus on the reactive organic compounds, and we therefore do not have tentative identification of the other organics present in the samples. Since we have not chemically characterized these species, we cannot hypothesize why these ions are not oxidized by OH. The organic species with "reactant-like" trends make up about 30% of the ESI-MS total ion abundance in the analyzed samples. We now say this in section 3.1:

*"Together, these ions make up roughly 30% of the positive mode total ion abundance in the experiment samples."*

It is possible that some compounds that appear to be unreactive are instead both formed and reacted by the complex mix in such a way that a trend cannot be observed.

**3) While there are some comparisons of mist-chamber species (collected WSOC) and CIMS (gas-phase compounds), this is only for a few select compounds. A more comprehensive comparison of the data from the two techniques is an important and necessary test of the hypothesis that these mist chambers are collecting WSOGs from the gas phase. For example, are there any WSOGs (other than ISOPOOH) measured by the CIMS (or PTRMS, etc.) that are not measured in the mist chamber? Conversely, what could explain any compounds measured in the mist chamber but not by the CIMS (e.g., m/z 125)? Finally, for the ions that are measured by both techniques, the ion intensities should be compared in all cases. (Figure S2 shows only a subset.)**

Yes, there are masses identified by the CIMS and not the mist chamber and vice versa. We have only plotted masses that were reported by both. The comparisons with the

CIMS are less useful that we had hoped. A major advantage of the CIMS is the much better time resolution. However, the CIMS is not able to distinguish between multiple compounds with the same elemental formula and each CIMS ionization reagents is sensitive to some compounds and not to others (Lopez-Hilfiker et al., 2016). The advantage of the mist chamber is that it collects a much wider range of water-soluble organic compounds (WSOCs) and that samples are available for off-line analyses, such as FT-ICR MS-MS. The MS-MS fragmentation helps us to distinguish between compounds with the same elemental formula but different structures. We expect some overlap between species measured by the two methods, but we also expect differences, and just because the two methods measure compounds with the same elemental formula does not mean they are measuring the same compounds. For example, we expect to find several isoprene-derived compounds with the same elemental composition at SOAS, and the relative ionization efficiencies of these compounds using CIMS are very different. Thus, even if the mist chamber collected all these compounds perfectly and there were no losses during storage, we would not expect the correlation between the CIMS and mist chamber ESI-MS signals to be strong. This is due to the different ionization methods used. All the species that are detected in the mist chamber using the ESI-MS are not necessarily detected using the CIMS. Additionally, collection/storage losses could mean that the CIMS detects some compounds that the mist chambers do not.

To make this clearer, we now have placed the CIMS comparisons under a separate header called "***Comparison of CIMS and ESI-MS results,***" and moved this section to SI. We added
*"The sensitivity of the I⁻ CIMS depends strongly on compound structure. Thus, we expect to find compounds in the mist chamber samples that are not detected by I⁻ CIMS, and it is likely that real-time CIMS analysis facilitates measurement of some species that we will not be able to detect in integrated mist chamber samples."*

***4) Throughout the paper, oxidation results (sample + H2O2 + UV) are compared to results from two blanks: sample + H2O2 and sample + UV. A third one that is at least as important is oxidation (UV+H2O2) of the sample-blank (water from the mist chamber that was exposed to zero air rather than ambient air, ideally sent through the same sampling setup). This is necessary for assessing whether any organic species were introduced by the sample lines, glassware, or sample handling.***

The reviewer's suggestion to sample "zero air" is challenging, because of the high flow rates. However, in addition to sample + H2O2 and sample + UV, we conducted a field water + H2O2 + UV control experiment. The field water blanks were handled, transported and stored identically to samples. The OH-reactive water-soluble ions identified in the sample + OH experiments were not found in the field water + H2O2 + UV control experiment.

We added to section 3.1: *"In field water + OH control experiments, these ions were not seen, confirming that they are not contaminants."*

***Other Comments:***

*Lines 68-79: This sentence would be more useful if the references were put throughout the sentence (connecting studies with individual features of SOA) rather than all at the end. This is particularly important because of the argument that there are "SOA discrepancies" involving high-MW, sulfur-containing, and nitrogen-containing species. I haven't read all 30 cited references closely, but I'm unaware of any such discrepancies (presumably between measurements and models?), since those organic types are usually not followed explicitly in models, nor are measured routinely in the ambient atmosphere.*

We have clarified this sentence in the main text to explain that the discrepancies refer to key atmospheric observations not explained by semi-volatile partitioning theory, as follows:
"Inclusion of aqueous chemistry of clouds, fogs, and wet aerosols in models and experiments helps to explain discrepancies *in atmospheric observations of SOA that aren't explained by semi-volatile partitioning theory,…*"
This is a good suggestion by the reviewer to separate out the references, but since quite a few of the references would include multiple features highlighted in these lines, we have opted to leave it as is. Following is an abbreviated list of some references pointing to the features:

- high atmospheric O/C ratios: Volkamer et al., 2007; Carlton et al., 2006
- enrichment of organic aerosol aloft: Heald et al., 2006; Sorooshian et al., 2010
- formation of oxalate, sulfur- and nitrogen-containing organics and high molecular weight compounds: Kawamura and Ikushima, 1993; Kawamura et al., 1996; Crahan et al., 2004; Kalberer et al., 2004; Herrmann et al., 2005; Altieri et al., 2006; Galloway et al., 2009

*Line 208: Glyoxal is just one of several standard compounds added. Did all the standards show a similar behavior?*

Other standards also showed a behavior similar to glyoxal. Glyoxal variance as mentioned in the text is <6%; for the other two standards, methylglyoxal and glycolaldehyde, the variance is 7% and 4%, respectively.

This has been added.

*238-240, 263-264: given that an FTICR is used in this study, these discussions of the implications of unit-mass resolution m/z values seem unnecessary.*

As per the reviewer recommendation, we have removed this discussion from the text.

*251, Fig. 1, and elsewhere: What is the approximate aqueous OH concentration in these experiments?*

We provided the OH production rate from hydrogen peroxide photolysis. In order to estimate an aqueous OH concentration in these experiments, we would need to be able to

model the chemistry occurring in the reaction vessel. Tan et al. conducted OH oxidation experiments of glyoxal at cloud water condition and modeled the reaction vessel chemistry to predict the OH concentration (Tan et al., 2009). If we assume that all the water-soluble organic gases collected in the mist chamber behave similarly to glyoxal, since our TOC values for the mist chamber are within the range of the experiments conducted by Tan et al., and OH production rate was similar as well, we estimate OH concentrations to be similar to those of Tan et al., i.e. $3 \times 10^{-12}$ M to $6 \times 10^{-12}$ M.

We now say this in section 2.2:

*"Note, while we can calculate the OH production rate from hydrogen peroxide photolysis ($1.25 \times 10^{-2}$ $\mu$M [OH] $s^{-1}$), the concentration of OH in the reaction vessel depends also on the reactivity of the organics. If the WSOG mix behaved similarly to glyoxal, OH concentrations would be on the order of $10^{-12}$ M (i.e., similar to Tan et al)"*

**269-273: I think the authors mean Figure 4 instead of Figure 3 here. This whole section could be removed, since it introduces a lot of information and raises questions that are not addressed until later in the paper.**

We thank the reviewer and have corrected the text to read Figure 4 instead of Figure 3 and have removed part of the text as suggested.

**Figure 2: the positive ion (Na+) should be given here, as it is in Figure 4.**

We have included Na+ in Figure 2.

**Lines 284-285, 324-325; Figure 3: This chemistry all involves the formation of diols from C-C double bonds. However diols are not generally considered to be alkene oxidation products. The main routes for formation of diols are when RO2+RO2 reactions dominate (e.g., Ruppert and Becker 2000, Atmos. Environment, 34, 1529-1542) or in the oxidation of conjugated dienes (leading to the formation of isoprene tetrols). What is the mechanism proposed here? Similarly, have these diol species been measured in any laboratory product studies?**

The putative dihydrodiol can form via an epoxide by a reaction analogous to that published for the formation of dihydroxyisopentanol (DHIP) from 2-methyl-3-buten-2-ol (MBO) (Zhang et al., 2014). In water, the epoxide hydrolyzes to the corresponding dihydrodiol. The formation of 2,3-dihydroxypentanal can thus be explained by the scheme below.

*Scheme: Formation of 2,3-dihydroxypentanal from pent-2,3-enal*

1,5-H shift

H₂O

+ HO₂

or

1,5-H shift

H₂O

+ HO₂

DHP, in fact, can be cited an example of a diol generated from an alkene measured both in laboratory and field studies.

***Line 288: These aren't "mechanisms" but simply measured compounds mapped back to potential reactants.***

"Mechanisms" has been removed from the text.

***Lines 290-319, Figure 4: When MeOH is added to the sample (for ESI analysis), might there be an exchange between –OH and –OCH3 groups? If so, this species might simply be from an isoprene tetrol.***

Unactivated alcohol OH, such as the hydroxyl groups of 2-methyltetrol will not exchange with methanol in solution to give a methoxy substituent, unlike the situation for carboxylic acids where formation of the corresponding methyl ester is possible. The exchange suggested by the reviewer has not been reported in any of the numerous LC/ESI-MS analyses of 2-methyltetrols in isoprene SOA. The $MS^2$ data are best explained by formation of a hemiacetal via addition of methanol to an aldehyde, which is a well-known reaction.

***Figure 5: I would think some of these structures could undergo fragmentation during MS-MS analysis. As stated in line 315, O-O bonds should break in this case.***

Since Figure 5 is purely speculative and we do not have supportive fragmentation data to back up any of the proposed structures, we have decided that it would be best to remove this figure and the related text from the manuscript and avoid any confusion or

misinformation for the readers.

*Figure 5: The structures given for m/z 125 are quite non-polar, with only one functional group in an eight-carbon molecule. Are they sufficiently water-soluble that they would be expected to be trapped in the mist chamber (or in cloud/fog droplets in the atmosphere)?*

As stated in the previous comment, this figure has been removed from the text.

*297-319: The argument in this paragraph was hard to follow. First it is stated that the CIMS and ESI measurements could "be the same compound". But then it's argued that CIMS is measuring IEPOX/ISOPOOH, which is not measured in the mist-chamber samples by ESI. But yet this ion is given as one of the CIMS-ESI comparisons in S2, suggesting the authors are arguing the species are in fact the same. This needs to be clarified.*

We have edited this paragraph for clarity. Two or more compounds are detected with the same elemental composition but with different sensitivities in the CIMS. Thus the CIMS signal might contain the compound we tentatively identified, but we cannot be sure. Also, the ion discussed here is a fragment and not a parent ion.

*Line 326: the structure given in Fig. 4 is not consistent with isoprene oxidation, since it has two methyl groups. The only plausible isoprene product (with one methyl group) of that formula that I can think of is IEPOX.*

The reviewer is correct and we have removed this statement from the draft.

*Figure S2: Plots need axis labels and units.*

Units are normalized ion abundance. We have included the label for sampling dates as per the reviewer's suggestion and added the following text to the figure caption:
*"Normalized ion abundance from ESI-MS and HRToF-CIMS for compounds with the same elemental formula."*

*Lines 381-382: This observation may simply be a result of a historical lack of good techniques for measuring gas-phase pyruvate and oxalate (i.e., pyruvic acid and oxalic acid). These compounds can now be measured routinely with CIMS (with acetate ionization), so these sorts of comparisons haven't really been able to be made well until now.*

The reviewer correctly points out that CIMS measurements can help determine the gas to particle ratio of oxalic acid. However, similar high quality gas-particle partitioning measurements have been conducted earlier using other techniques such as those used by Martinelango et al. during the Bay Region Atmospheric Chemistry Experiment (BRACE) (Martinelango et al., 2007). Their measurements show that particle-phase oxalate concentrations are much greater than gas-phase concentrations. Additionally, field

measurements of dicarboxylic acids in atmospheric particles in Hong Kong find only about 6-12 % of total oxalic acid in the gas-phase (Yao et al., 2002). We reference these papers.

*Lines 426: What were the "several" isoprene products? I think only one (C5H10O3) is discussed in the paper (but see my comments above).*

As per the reviewer's suggestion, we have changed this line to read:
*"...and some are tentatively gas-phase oxidation products of green leaf volatiles.."*

*Overall: From the concentrations of oxalate/pyruvate formation, this work should give a rough upper limit for the amount of aqueous SOA that can be formed from cloud/fog processing in a given sample (assuming collection of all gas-phase WSOC). What is this value?*

We agree with the reviewer that it would be helpful to provide an upper limit for the amount of aqueous SOA that can be formed from cloud/fog processing in a given sample. In doing so, we would have to make quite a few assumptions such as collection of all gas-phase WSOC (as the reviewer suggests), conversion of all SOA to oxalate/pyruvate etc. that could greatly distort the results and mislead the community, and hence we have opted to not provide such an estimate. A technique that could provide a better estimate would be to conduct droplet evaporation experiments on the collected samples.

**REVIEWER 2**

*The authors of this manuscript collected gas phase compounds during the SOAS measurement campaign and measured their reaction progress in the presence of OH to understand the aqueous phase chemistry that occurs as a result of gas-to-particle partitioning of these compounds. They measured loss rates of several oxidized compounds and formation rates of four organic acids as a result of reaction of these ambient compounds with OH and monitored the formation of highly oxidized oxalate, pyruvate, acetate, and glycolate over time. This indicates that the aqueous phase oxidation of water soluble organic compounds may be very important for the formation of these compounds, possibly followed by repartitioning back to the gas phase after oxidation. This study is important for the atmospheric chemistry because it uses ambient compounds as precursors for oxidation, instead of single "proxy" compounds or simple mixtures of such. The authors were able to track both precursor and product compounds, which could lead to mechanistic conclusions about the formation of organic acids in atmospheric waters. However, more details are needed about both the sampling method (e.g., recovery of compounds through the mist chambers and instrumental precision, see comments below) and about other compounds besides those discussed here. This will give the reader a better sense of the significance organic acid formation from the compounds discussed here.*

*The five masses that exhibited "reactant-like trends" are discussed in detail. Are these the only compounds observed in the initial mixtures? If not, were there others that*

*reacted with OH or that stayed constant over the course of the experiment? It seems unlikely that these are the only compounds detected, and more detail would be appreciated to give the reader a sense of the concentrations of these compounds as compared to others. How significant were these five compounds in terms of the percentage of organic matter? Why wouldn't other compounds react with OH? How likely is it that compounds that are not detected contribute to organic acid formation? If these are the only compounds that displayed "reactant-like trends," then this will allow for the conclusion that oxalate, pyruvate, acetate, and glycolate are directly produced from these compounds and not from others. Otherwise, this argument is hard to make. A similar comment can be made for the oxidized species formed from the OH reaction. Only four organic acid products are mentioned in the manuscript. Were others observed? If not, is this expected? The reaction of OH with organics is likely to produce these small organic acids eventually, but I would expect to see other acids formed as intermediates in this process as well.*

*Discussion of the significance of these compounds in WSOC would strengthen this manuscript. As is, the authors make some interesting conclusions about the formation of aqueous SOA formation, but it is difficult to determine the magnitude of their contributions to aqueous phase chemistry. An upper bound estimate of this contribution might also be interesting to see.*

As the reviewer suggests, we do observe other organic species (not discussed in the text) that do not react with OH. Since the focus of our investigative effort and chemical characterization was on compounds that reacted, we do not discuss "non-reacting" ions. In fact, FT-ICR-MS time is expensive and analyses are time consuming. Thus we focused our instrument time on the reactive compounds. Since we have not chemically characterized these species, we cannot hypothesize why these ions are not oxidized by OH. The organic species with "reactant-like" trends account for about 30% of the total ion abundance observed in the mist chambers. This is now indicated in the text:

*"Together, these ions make up roughly 30% of the positive mode total ion abundance in the experiment samples"*

The organic acids observed in this study are formed as intermediates or products and as the reviewer says, OH oxidation will lead to their formation. It is certainly possible that other acids could be formed.

We agree with the reviewer that it would be helpful to provide an upper limit for the amount of aqueous SOA that can be formed from cloud/fog processing in a given sample. In doing so, we would have to make quite a few assumptions such as collection of all gas-phase WSOC, conversion of all SOA to oxalate/pyruvate etc. that could greatly distort the results and mislead the community, and hence we have opted to not provide such an estimate. A technique that could provide a better estimate would be to conduct droplet evaporation experiments on the collected samples.

**Specific comments**

*The authors state in the methods section that intensive days were chosen because of high concentration predictions by NCAR and that during those days, they observed higher than usual TOC content in general. However, in examining Supplementary Table S1, it seems that the days marked "intensive" are not much higher than others, if at all. The range given in the text (92-179 μM-C) does not match the range of TOC on intensive days. Can this be clarified? Does this range only represent the days that were used in further measurements?*

We apologize for this confusion. To clarify, we primarily focused on the intensive days that had the highest TOC contents; primarily the first three intensive periods highlighted in the table. Besides these days, we also chose to pick July $20^{th}$ and $21^{st}$ to perform experiments due to the high TOC values on these days. We have clarified this in the text:

*"In general, mist chamber samples on intensive sampling days had higher organic content (TOC = 92-179 μM-C), and hence we focused our experiments on those days and included two additional days from the non-intensive period that had high TOC values (Table 1)."*

*Line 208: Why is glyoxal the only standard compared across all analysis days? Were the variabilities of the other standards on a similar scale? If so, this should be stated here.*

Similar to glyoxal, the other standards also showed a similar behavior. Glyoxal variance as mentioned in the text is <6%; for the other two standards, methylglyoxal and glycolaldehyde, the variance is 7% and 4%, respectively. This is now noted in the text.

*Line 215-219: Limit of detection and precision information is given here for oxalate, but this is not the only organic acid measured by IC. Can all the acids studied be considered to be similar to this or do they vary significantly?*

Yes, we typically find the precision and detection limits of oxalate, pyruvate, and acetate to be similar. However, in this work we used ESI-MS for pyruvate.

*In Figure 2, m/z 187 is actually the sodium cluster, which is not shown in the figure. However, Figure 4 makes it clear that this is the case. These should be changed to be consistent with each other.*

We have included Na+ in Figure 2.

*A predicted structure for m/z 187 is shown in Figure 2, along with the predicted gas- and aqueous- phase structures. However, the observed mass contains an extra methoxy group. From previous comments on the methodology and the discussion that follows about the compound at m/z 173, I assume this is a result of hydration by methanol in the FT-ICR-MS. However, this is not stated until after the discussion of m/z 187, and should be explained the first time it is seen.*

We thank the reviewer for this suggestion and have included this information in the text for m/z 187 as follows:

"*The positive ion at m/z 187 is seen hydrated with methanol and upon fragmentation, methanol is lost and the daughter ion peak is observed at m/z 155.0680, corresponding to the molecular formula, $C_6H_{12}O_3$.*"

**In line 319 and again in lines 360-364, the authors state that they expect IEPOX, ISOPOOH, and glyoxal were lost in their experiments during sampling or storage. Have any tests been done to quantify losses of other compounds during sampling or storage? If these compounds were lost, it seems likely that there are others that are either lost or not fully recovered. Were any recovery studies done with known amounts of standards instead of spiking the samples just before analysis?**

We have not performed recovery studies, although this would be a good addition to our future work. However, we have now added a section called "Methodological limitations" of the work that focuses on issues such as losses:

"*It should be noted that WSOGs can be lost during sampling and storage through: 1) losses in tubing and by adsorption to the QFF during collection, 2) reactions in the mist chamber during collection with water-soluble ambient oxidants capable of penetrating the inlet (e.g., ozone), and 3) losses during storage post collection. The QFF removes particles upstream of the mist chambers. In the early stages of sampling, on the clean filter, adsorption of gases on the filter will reduce the concentrations of gases sampled by the mist chamber until these gases reach gas phase – adsorbed phase equilibrium. Using glyoxal as a WSOG-surrogate and the work of (Mader and Pankow, 2001) we predict that the measured WSOG in the mist chamber will be depleted for less than 2% of our sampling time (after $<0.1 \ m^3$) due to loss to the filter. Thus, we expect water-soluble gases to penetrate through the QFF very efficiently for collection in the mist chamber water. Losses to Teflon inlets and chamber walls (Krechmer et al., 2016) can be significant and variable and may reduce the number of species we are able to collect and identify in this work. While OH radicals are unlikely to penetrate the inlet, ozone might. Thus some ozone could be scrubbed by the mist chambers and could result in oxidation of some WSOGs during collection. Though many organics are stable when stored frozen in water, IEPOX does not survive extended storage in water (confirmed with our organic synthesis collaborator). We expect this to be the case for ISOPOOH also. ISOPOOH is an OH oxidation product of isoprene, which is further oxidized by OH under low-NO conditions to form isomeric IEPOX (Paulot et al., 2009). Both IEPOX and ISOPOOH are prevalent at the SOAS ground site due to the abundance of isoprene emissions in this forested region. These losses constitute the major limitation of the work in that they restrict the number of reactive water-soluble compounds we are able to identify.*"

**In Figure 5, for m/z 125, how likely is it that the first structure shown (the aldehyde) is detected as such and not hydrated in the aqueous mixture? Is this a likely structure? The authors also state in the discussion of this figure that gas phase compounds are**

*seen at m/z 129 and 143. However, it seems unlikely that these compounds are the polyols found in Figure 5. Are there any compounds with those masses that may be found in the same form in both the gas- and aqueous- phase?*

Since Figure 5 is purely speculative and we do not have supportive fragmentation data to back up any of the proposed structures, we have decided that it would be best to remove this figure and the related text from the manuscript and avoid any confusion or misinformation for the readers.

*In the Figure 6 caption, special mention is made of the fact that oxalate is present in all samples. Is this not true for pyruvate as well?*

This is true for both pyruvate and oxalate and have corrected this in the figure caption.

*I'm not really sure why Supplementary Figure S3 is not included in the main portion of the manuscript. It is discussed in the main text and seems to be important to the overall conclusions of the paper. It is also hard to follow, mainly because there are so many points. It would be easier to see the trends if a general trend line was added as in Figure 6.*
*Acetate and glycolate are found in some samples, but in varying concentrations. Did the authors see any trends that might explain their formation on some days and not others? Also related, if acetate and glycolate co-elute on the IC, how can the statement be made that "Acetate formation is seen on some but not all days" when any signal seen is due to the combination of both? Was acetate also measured via another method? There seems to be a lack of discussion about the glycolate present. In the discussion of these results (lines 375-380), acetate is mentioned but not glycolate. Is this because it is expected that most of this combined signal is acetate or because glycolate is not expected to be an important oxalate precursor in these ambient mixtures?*

Supplementary Figure S3 shows acetate and glycolate production during sample oxidation experiments. As shown in the figure, there are two experimental days where there is a higher production of these compounds, but there are no trends (e.g. relation to high initial sample TOC values) that we have observed to explain their higher formation on some days versus others. On some of the days with lower production of acetate/glycolate, the trend lines would be similar to the control experiments, not allowing us to conclusively assert that they are formed on all days, and hence we have not included this figure in the main text. The reviewer is correct that acetate and glycolate co-elute and we have no other way of distinguishing between the two. We have corrected the statement to include glycolate:

"*Acetate/ glycolate formation is seen on some but not all days*" and include glycolate in the results and discussion.

*Were any measurements of oxalate in the particle phase taken at SOAS? in lines 421-423, the authors state that based on their conclusions, it is unlikely that oxalate will be*

*present in the particle phase, but it would be interesting to test this.*

They do not seem to be published yet.

*Technical Corrections:*

*Page 6, line 140: "ml" should be "mL"*
*Page 7, line 159: "Henry's law" should be "Henry's Law"*
*Page 12, line 270: Should this refer to Figures 2 and 4 instead of 2 and 3?*
*Page 13, line 288: Figure 3 does not show a mechanism, but only initial and final structures.*
*Page 14, line 327: It is unclear to me why the word "these" is italicized.*
*Page 16, line 363: There is an extra parenthesis at the end of the paragraph.*
*Supplementary Figure S2: This figure is missing axes labels.*
*Supplementary Figure S3 caption, line 2: "co-elude" should be "co-elute"*

We thank the reviewer for these technical corrections and have addressed them in the final draft.

**References**

Krechmer, J. E., Pagonis, D., Ziemann, P. J., and Jimenez, J. L.: Quantification of Gas-Wall Partitioning in Teflon Environmental Chambers Using Rapid Bursts of Low-Volatility Oxidized Species Generated in Situ, Environmental Science & Technology, 50, 5757-5765, 10.1021/acs.est.6b00606, 2016.

Lopez-Hilfiker, F. D., Iyer, S., Mohr, C., Lee, B. H., D'Ambro, E. L., Kurtén, T., and Thornton, J. A.: Constraining the sensitivity of iodide adduct chemical ionization mass spectrometry to multifunctional organic molecules using the collision limit and thermodynamic stability of iodide ion adducts, Atmos. Meas. Tech., 9, 1505-1512, 10.5194/amt-9-1505-2016, 2016.

Mader, B. T., and Pankow, J. F.: Gas/Solid Partitioning of Semivolatile Organic Compounds (SOCs) to Air Filters. 3. An Analysis of Gas Adsorption Artifacts in Measurements of Atmospheric SOCs and Organic Carbon (OC) When Using Teflon Membrane Filters and Quartz Fiber Filters, Environmental Science & Technology, 35, 3422-3432, 10.1021/es0015951, 2001.

Martinelango, P. K., Dasgupta, P. K., and Al-Horr, R. S.: Atmospheric production of oxalic acid/oxalate and nitric acid/nitrate in the Tampa Bay airshed: Parallel pathways,

Atmospheric Environment, 41, 4258-4269, http://dx.doi.org/10.1016/j.atmosenv.2006.05.085, 2007.

Paulot, F., Crounse, J. D., Kjaergaard, H. G., Kurten, A., St Clair, J. M., Seinfeld, J. H., and Wennberg, P. O.: Unexpected Epoxide Formation in the Gas-Phase Photooxidation of Isoprene, Science, 325, 730-733, DOI 10.1126/science.1172910, 2009.

Tan, Y., Perri, M. J., Seitzinger, S. P., and Turpin, B. J.: Effects of Precursor Concentration and Acidic Sulfate in Aqueous Glyoxal−OH Radical Oxidation and Implications for Secondary Organic Aerosol, Environmental Science & Technology, 43, 8105-8112, 10.1021/es901742f, 2009.

Yao, X., Fang, M., and Chan, C. K.: Size distributions and formation of dicarboxylic acids in atmospheric particles, Atmospheric Environment, 36, 2099-2107, http://dx.doi.org/10.1016/S1352-2310(02)00230-3, 2002.

Zhang, H., Zhang, Z., Cui, T., Lin, Y.-H., Bhathela, N. A., Ortega, J., Worton, D. R., Goldstein, A. H., Guenther, A., Jimenez, J. L., Gold, A., and Surratt, J. D.: Secondary Organic Aerosol Formation via 2-Methyl-3-buten-2-ol Photooxidation: Evidence of Acid-Catalyzed Reactive Uptake of Epoxides, Environmental Science & Technology Letters, 1, 242-247, 10.1021/ez500055f, 2014.

---

## Author Response (AR1)

**REVIEWER 1**

Review of "Identifying precursors and aqueous organic aerosol formation pathways during the SOAS campaign" by Sareen et al.

The paper describes measurements of the oxidative chemistry of atmospheric watersoluble organic compounds (WSOC). Mist chamber samples were taken during the SOAS campaign, for the collection of gas-phase WSOC. Collected organic species were then oxidized offline by OH radicals (generated by addition of H2O2 followed by UV irradiation), and oxidation chemistry was tracked using ESI-MS and IC. A small number of organic species were found to decrease with OH oxidation, and a few oxidation products were found to be formed. These results are interpreted in terms of the formation of secondary organic aerosol (SOA) within cloud or fog droplets.

This is useful study on an important topic in atmospheric chemistry. The overall approach (collection and aqueous-phase oxidation of ambient WSOC) is a novel one, and has the potential to provide insights into the role of aqueous-phase oxidation in SOA formation. However, the actual measurements described in this manuscript raise a number of questions about the relationships of the organics collected/measured to those in the ambient atmosphere – examples include the fraction of ambient gas-phase WSOG that are actually collected, the possibility of loss of organic species during sampling, and the potential role of background organic species. These concerns are described below, and need to be addressed if this work is to be published in ACP.

We thank the reviewer for their helpful feedback and suggestions. We address each point in order below.

1) The authors point out that there are some large discrepancies between in-situ measurements of gas-phase water-soluble organic gases (e.g., ISOPOOH, glyoxal) and the species measured within the mist chambers. This is attributed to "loss during sampling or storage" (line 319), with the possibility that they may have undergone oxidation within the water (line 360). Since such losses have major implications for the generalization of results, these need to be discussed more thoroughly. First, irreversible loss to the sample lines/inlets can be a major sink for IVOCs, so this needs to be considered. What was the sampling scheme used? (The length, diameter, and material used for the sample tubing needs to be given.) What is the sampling efficiency of water-soluble standards sent through this sampling setup? This is briefly touched on in the paper, in a discussion of glyoxal loss to the particle filter (lines 134-136), but this is based on rough calculations and not actual measurements, and doesn't consider the role of losses to tubing.

Further, if oxidation within the collected (non-irradiated) sample is occurring (lines 360-363), there is some important chemistry here that needs to be discussed. The authors appear to be arguing that glyoxal and ISOPOOH react with in-situ, non-OH oxidants (H2O2, etc). Is this consistent with previous studies, and the existing literature? Moreover such behavior would have to be different from that of compounds focused on in this study: they would have to be resistant to oxidation by any oxidants

collected within the mist-chamber samples, but still reactive with OH. Is this the argument being made here? If so, what are the implications for atmospheric oxidation of these various species?

These suggestions to better characterize and discuss potential losses are helpful.

The length, diameter and material of the inlet tubing has been added to the first paragraph of methods section 2.1, line 132:

"Samples were collected from June 1 – July 14, 2013 from 1 m above the trailer roof through a 1.3 cm OD PTFE inlet (approximately 1.7 m in length)."

The reviewers' comments have been helpful in clarifying our thoughts on losses.

There are three major ways to lose water-soluble organic gases during sampling/storage:

- 1. Losses in tubing. Recent work by Krechmer et al highlight the uncertainties associated with quantifying the loss of gas-phase organic compounds to Teflon (Krechmer et al., 2016). This limitation is now acknowledged.
- 2. Ozone could also be collected during sampling and could react with certain organic gases during mist chamber collection.
- 3. Storage and sample handling. As described in the text, at the end of each collection day, the samples were separated into experimental sized aliquots and frozen immediately. They were shipped overnight with ice packs to the laboratory at Rutgers, where they were placed in the freezer at -20°C upon arrival. Before an experiment the sample was thawed at room temperature.

IEPOX does not survive extended storage in water (confirmed with our organic synthesis collaborator). We expect that this is also the case for ISOPOOH. This explains why we did not see IEPOX and ISOPOOH in our collected samples.

We have added a section called "Methodological Limitations" in the Results & Discussion section (line 300) to address losses. We had already addressed losses to the quartz fiber filter, so that text has now been moved down to the new paragraph, which reads:

"However, WSOGs can be lost during sampling and storage through: 1) losses in tubing and by adsorption to the QFF during collection, 2) reactions in the mist chamber during collection with water-soluble ambient oxidants capable of penetrating the inlet (e.g., ozone), and 3) losses during storage post collection. The QFF removes particles upstream of the mist chambers. In the early stages of sampling, on the clean filter, adsorption of gases on the filter will reduce the concentrations of gases sampled by the mist chamber until these gases reach gas phase – adsorbed phase equilibrium. Using glyoxal as a WSOG-surrogate and the work of (Mader and Pankow, 2001) we predict that the measured WSOG in the mist chamber will be depleted for less than 2% of our sampling time (after <0.1 m3) due to loss to the filter. Thus, we expect water-soluble gases to penetrate through the QFF very efficiently for collection in the mist chamber

water. Losses to Teflon inlets and chamber walls (Krechmer et al., 2016) can be significant and variable and may reduce the number of species we are able to collect and identify in this work. While OH radicals are unlikely to penetrate the inlet, ozone might. Thus some ozone could be scrubbed by the mist chambers and could result in oxidation of some WSOGs during collection.

Though many organics are stable when stored frozen in water, IEPOX does not survive extended storage in water (confirmed with our organic synthesis collaborator). We expect this to be the case for ISOPOOH also. ISOPOOH is an OH oxidation product of isoprene, which is further oxidized by OH under low-NO conditions to form isomeric IEPOX (Paulot et al., 2009). IEPOX and ISOPOOH were present in the gas-phase during the SOAS campaign (Nguyen et al., 2015). They have relatively high Henry's Law constants (i.e.,  $(H_{L,IEPOX}=2.7 \times 10^6 \text{ M atm}^{-1})$ . IEPOX was readily detected in ambient samples spiked with 3000  $\mu$ M, 300  $\mu$ M, and 30  $\mu$ M of IEPOX, indicating that it can be ionized in our sample matrix. (Authentic trans- $\beta$ -IEPOX, which is the predominant isomer of IEPOX, was synthesized for this purpose (Zhang et al., 2012).) However, it was not found in our ambient samples since it is not stable when stored in water."

**2) The paper focuses entirely with organic species that exhibit "reactant-like trends". However there also should be some discussion of ions that do not exhibit such trends, if any. Were any of these observed? If so, what fraction of total WSOC signal do they make up? What are their formulas and characteristics? (Why are they not oxidized by OH?)**

As the reviewer suggests, we do observe other organic species (not discussed in the text) that do not react with OH. Woods Hole high-resolution instrument time is expensive and valuable. We used our instrument time to focus on the reactive organic compounds, and we therefore do not have tentative identification of the other organics present in the samples. Since we have not chemically characterized these species, we cannot hypothesize why these ions are not oxidized by OH. The organic species with "reactant-like" trends make up about 30% of the ESI-MS total ion abundance in the analyzed samples. We now say this in section 3.1, line 346:

**"Together, the ions discussed herein account for 30% of the total ion abundance."**

It is possible that some compounds that appear to be unreactive are instead both formed and reacted by the complex mix in such a way that a trend cannot be observed.

3) While there are some comparisons of mist-chamber species (collected WSOC) and CIMS (gas-phase compounds), this is only for a few select compounds. A more comprehensive comparison of the data from the two techniques is an important and necessary test of the hypothesis that these mist chambers are collecting WSOGs from the gas phase. For example, are there any WSOGs (other than ISOPOOH) measured by the CIMS (or PTRMS, etc.) that are not measured in the mist chamber? Conversely, what could explain any compounds measured in the mist chamber but not by the CIMS (e.g., m/z 125)? Finally, for the ions that are measured by both techniques, the ion intensities should be compared in all cases. (Figure S2 shows only a subset.)

Yes, there are masses identified by the CIMS and not the mist chamber and vice versa. We have only plotted masses that were reported by both. The comparisons with the CIMS are less useful that we had hoped. A major advantage of the CIMS is the much better time resolution. However, the CIMS is not able to distinguish between multiple compounds with the same elemental formula and each CIMS ionization reagents is sensitive to some compounds and not to others (Lopez-Hilfiker et al., 2016). The advantage of the mist chamber is that it collects a much wider range of water-soluble organic compounds (WSOCs) and that samples are available for off-line analyses, such as FT-ICR MS-MS. The MS-MS fragmentation helps us to distinguish between compounds with the same elemental formula but different structures. We expect some overlap between species measured by the two methods, but we also expect differences, and just because the two methods measure compounds with the same elemental formula does not mean they are measuring the same compounds. For example, we expect to find several isoprene-derived compounds with the same elemental composition at SOAS, and the relative ionization efficiencies of these compounds using CIMS are very different. Thus, even if the mist chamber collected all these compounds perfectly and there were no losses during storage, we would not expect the correlation between the CIMS and mist chamber ESI-MS signals to be strong. This is due to the different ionization methods used. All the species that are detected in the mist chamber using the ESI-MS are not necessarily detected using the CIMS. Additionally, collection/storage losses could mean that the CIMS detects some compounds that the mist chambers do not.

To make this clearer, we now have placed the CIMS comparisons under a separate header called "*Comparison of CIMS and ESI-MS results*," and moved this section to SI. We added

"The sensitivity of the  $\Gamma$  CIMS depends strongly on compound structure. Thus, we expect to find compounds in the mist chamber samples that are not detected by  $\Gamma$  CIMS, and it is likely that real-time CIMS analysis facilitates measurement of some species that we will not be able to detect in integrated mist chamber samples."

4) Throughout the paper, oxidation results (sample + H2O2 + UV) are compared to results from two blanks: sample + H2O2 and sample + UV. A third one that is at least as important is oxidation (UV+H2O2) of the sample-blank (water from the mist chamber that was exposed to zero air rather than ambient air, ideally sent through the same sampling setup). This is necessary for assessing whether any organic species were introduced by the sample lines, glassware, or sample handling.

The reviewer's suggestion to sample "zero air" is challenging, because of the high flow rates. However, in addition to sample + H2O2 and sample + UV, we conducted a field water + H2O2 + UV control experiment. The field water blanks were handled, transported and stored identically to samples. The OH-reactive water-soluble ions identified in the sample + OH experiments were not found in the field water + H2O2 + UV control experiment.

We added to section 3.1, line 246:

"In control experiments where we generated OH radicals in field water blanks, these ions were not observed, confirming that they are not contaminants,"

**Other Comments:**

Lines 68-79: This sentence would be more useful if the references were put throughout the sentence (connecting studies with individual features of SOA) rather than all at the end. This is particularly important because of the argument that there are "SOA discrepancies" involving high-MW, sulfur-containing, and nitrogen-containing species. I haven't read all 30 cited references closely, but I'm unaware of any such discrepancies (presumably between measurements and models?), since those organic types are usually not followed explicitly in models, nor are measured routinely in the ambient atmosphere.

We have clarified this sentence in the main text to explain that the discrepancies refer to key atmospheric observations not explained by semi-volatile partitioning theory, as follows:

"Inclusion of aqueous chemistry of clouds, fogs, and wet aerosols in models and experiments helps to explain discrepancies *in atmospheric observations of SOA that aren't explained by semi-volatile partitioning theory*,..."

This is a good suggestion by the reviewer to separate out the references, but since quite a few of the references would include multiple features highlighted in these lines, we have opted to leave it as is. Following is an abbreviated list of some references pointing to the features:

- high atmospheric O/C ratios: Volkamer et al., 2007; Carlton et al., 2006
- enrichment of organic aerosol aloft: Heald et al., 2006; Sorooshian et al., 2010
- formation of oxalate, sulfur- and nitrogen-containing organics and high molecular weight compounds: Kawamura and Ikushima, 1993; Kawamura et al., 1996; Crahan et al., 2004; Kalberer et al., 2004; Herrmann et al., 2005; Altieri et al., 2006; Galloway et al., 2009

**Line 208: Glyoxal is just one of several standard compounds added. Did all the standards show a similar behavior?**

Other standards also showed a behavior similar to glyoxal. Glyoxal variance as mentioned in the text is

**these experiments?**

We provided the OH production rate from hydrogen peroxide photolysis. In order to estimate an aqueous OH concentration in these experiments, we would need to be able to model the chemistry occurring in the reaction vessel. Tan et al. conducted OH oxidation experiments of glyoxal at cloud water condition and modeled the reaction vessel chemistry to predict the OH concentration (Tan et al., 2009). If we assume that all the water-soluble organic gases collected in the mist chamber behave similarly to glyoxal, since our TOC values for the mist chamber are within the range of the experiments conducted by Tan et al., and OH production rate was similar as well, we estimate OH concentrations to be similar to those of Tan et al., i.e.  $3 \times 10^{-12}$  M to  $6 \times 10^{-12}$  M.

We now say this in section 2.2, line 168:

"While we can calculate the OH production rate from hydrogen peroxide photolysis  $(1.25 \times 10^{-2} \ \mu M \ [OH] \ s^{-1})$ , the concentration of OH in the reaction vessel depends also on the reactivity of the organics. If the WSOG mix behaves similarly to glyoxal, OH concentrations would be on the order of  $10^{-12} M$  (i.e., similar to Tan et al 2009)"

**269-273: I think the authors mean Figure 4 instead of Figure 3 here. This whole section could be removed, since it introduces a lot of information and raises questions that are not addressed until later in the paper.**

We thank the reviewer and have corrected the text to read Figure 4 instead of Figure 3 and have removed part of the text as suggested.

**Figure 2: the positive ion (Na+) should be given here, as it is in Figure 4.**

We have included Na+ in Figure 2.

Lines 284-285, 324-325; Figure 3: This chemistry all involves the formation of diols from C-C double bonds. However diols are not generally considered to be alkene oxidation products. The main routes for formation of diols are when RO2+RO2 reactions dominate (e.g., Ruppert and Becker 2000, Atmos. Environment, 34, 1529-1542) or in the oxidation of conjugated dienes (leading to the formation of isoprene tetrols). What is the mechanism proposed here? Similarly, have these diol species been measured in any laboratory product studies?

The putative dihydrodiol can form via an epoxide by a reaction analogous to that published for the formation of dihydroxyisopentanol (DHIP) from 2-methyl-3-buten-2-ol (MBO) (Zhang et al., 2014). In water, the epoxide hydrolyzes to the corresponding dihydrodiol. The formation of 2,3-dihydroxypentanal can thus be explained by the scheme below.

Scheme: Formation of 2,3-dihydroxypentanal from pent-2,3-enal

DHP, in fact, can be cited an example of a diol generated from an alkene measured both in laboratory and field studies.

**Line 288: These aren't "mechanisms" but simply measured compounds mapped back to potential reactants.**

"Mechanisms" has been removed from the text.

**Lines 290-319, Figure 4: When MeOH is added to the sample (for ESI analysis), might there be an exchange between -OH and -OCH3 groups? If so, this species might simply be from an isoprene tetrol.**

Unactivated alcohol OH, such as the hydroxyl groups of 2-methyltetrol will not exchange with methanol in solution to give a methoxy substituent, unlike the situation for carboxylic acids where formation of the corresponding methyl ester is possible. The exchange suggested by the reviewer has not been reported in any of the numerous LC/ESI-MS analyses of 2-methyltetrols in isoprene SOA. The MS2 data are best explained by formation of a hemiacetal via addition of methanol to an aldehyde, which is a well-known reaction.

**Figure 5: I would think some of these structures could undergo fragmentation during MS-MS analysis. As stated in line 315, O-O bonds should break in this case.**

Since Figure 5 is purely speculative and we do not have supportive fragmentation data to back up any of the proposed structures, we have decided that it would be best to remove this figure and the related text from the manuscript and avoid any confusion or

misinformation for the readers.

Figure 5: The structures given for m/z 125 are quite non-polar, with only one functional group in an eight-carbon molecule. Are they sufficiently water-soluble that they would be expected to be trapped in the mist chamber (or in cloud/fog droplets in the atmosphere)?

As stated in the previous comment, this figure has been removed from the text.

297-319: The argument in this paragraph was hard to follow. First it is stated that the CIMS and ESI measurements could "be the same compound". But then it's argued that CIMS is measuring IEPOX/ISOPOOH, which is not measured in the mist-chamber samples by ESI. But yet this ion is given as one of the CIMS-ESI comparisons in S2, suggesting the authors are arguing the species are in fact the same. This needs to be clarified.

We have edited this paragraph for clarity. Two or more compounds are detected with the same elemental composition but with different sensitivities in the CIMS. Thus the CIMS signal might contain the compound we tentatively identified, but we cannot be sure. Also, the ion discussed here is a fragment and not a parent ion.

Line 326: the structure given in Fig. 4 is not consistent with isoprene oxidation, since it has two methyl groups. The only plausible isoprene product (with one methyl group) of that formula that I can think of is IEPOX.

The reviewer is correct and we have removed this statement from the draft.

**Figure S2: Plots need axis labels and units.**

Units are normalized ion abundance. We have included the label for sampling dates as per the reviewer's suggestion and added the following text to the figure caption: *"Normalized ion abundance from ESI-MS and HRToF-CIMS for compounds with the same elemental formula."*

Lines 381-382: This observation may simply be a result of a historical lack of good techniques for measuring gas-phase pyruvate and oxalate (i.e., pyruvic acid and oxalic acid). These compounds can now be measured routinely with CIMS (with acetate ionization), so these sorts of comparisons haven't really been able to be made well until now.

The reviewer correctly points out that CIMS measurements can help determine the gas to particle ratio of oxalic acid. However, similar high quality gas-particle partitioning measurements have been conducted earlier using other techniques such as those used by Martinelango et al. during the Bay Region Atmospheric Chemistry Experiment (BRACE) (Martinelango et al., 2007). Their measurements show that particle-phase oxalate concentrations are much greater than gas-phase concentrations. Additionally, field

measurements of dicarboxylic acids in atmospheric particles in Hong Kong find only about 6-12 % of total oxalic acid in the gas-phase (Yao et al., 2002). We reference these papers.

**Lines 426: What were the "several" isoprene products? I think only one (C5H10O3) is discussed in the paper (but see my comments above).**

As per the reviewer's suggestion, we have changed line 415 to read: "...and some are tentatively gas-phase oxidation products of green leaf volatiles.."

Overall: From the concentrations of oxalate/pyruvate formation, this work should give a rough upper limit for the amount of aqueous SOA that can be formed from cloud/fog processing in a given sample (assuming collection of all gas-phase WSOC). What is this value?

We agree with the reviewer that it would be helpful to provide an upper limit for the amount of aqueous SOA that can be formed from cloud/fog processing in a given sample. In doing so, we would have to make quite a few assumptions such as collection of all gasphase WSOC (as the reviewer suggests), conversion of all SOA to oxalate/pyruvate etc. that could greatly distort the results and mislead the community, and hence we have opted to not provide such an estimate. A technique that could provide a better estimate would be to conduct droplet evaporation experiments on the collected samples.

**REVIEWER 2**

The authors of this manuscript collected gas phase compounds during the SOAS measurement campaign and measured their reaction progress in the presence of OH to understand the aqueous phase chemistry that occurs as a result of gas-to-particle partitioning of these compounds. They measured loss rates of several oxidized compounds and formation rates of four organic acids as a result of reaction of these ambient compounds with OH and monitored the formation of highly oxidized oxalate, pyruvate, acetate, and glycolate over time. This indicates that the aqueous phase oxidation of water soluble organic compounds may be very important for the formation of these compounds, possibly followed by repartitioning back to the gas phase after oxidation. This study is important for the atmospheric chemistry because it uses ambient compounds as precursors for oxidation, instead of single "proxy" compounds or simple mixtures of such. The authors were able to track both precursor and product compounds, which could lead to mechanistic conclusions about the formation of organic acids in atmospheric waters. However, more details are needed about both the sampling method (e.g., recovery of compounds through the mist chambers and instrumental precision, see comments below) and about other compounds besides those discussed here. This will give the reader a better sense of the significance organic acid formation from the compounds discussed here.

The five masses that exhibited "reactant-like trends" are discussed in detail. Are these the only compounds observed in the initial mixtures? If not, were there others that

reacted with OH or that stayed constant over the course of the experiment? It seems unlikely that these are the only compounds detected, and more detail would be appreciated to give the reader a sense of the concentrations of these compounds as compared to others. How significant were these five compounds in terms of the percentage of organic matter? Why wouldn't other compounds react with OH? How likely is it that compounds that are not detected contribute to organic acid formation? If these are the only compounds that displayed "reactant-like trends," then this will allow for the conclusion that oxalate, pyruvate, acetate, and glycolate are directly produced from these compounds and not from others. Otherwise, this argument is hard to make. A similar comment can be made for the oxidized species formed from the OH reaction. Only four organic acid products are mentioned in the manuscript. Were others observed? If not, is this expected? The reaction of OH with organics is likely to produce these small organic acids eventually, but I would expect to see other acids formed as intermediates in this process as well.

Discussion of the significance of these compounds in WSOC would strengthen this manuscript. As is, the authors make some interesting conclusions about the formation of aqueous SOA formation, but it is difficult to determine the magnitude of their contributions to aqueous phase chemistry. An upper bound estimate of this contribution might also be interesting to see.

As the reviewer suggests, we do observe other organic species (not discussed in the text) that do not react with OH. Since the focus of our investigative effort and chemical characterization was on compounds that reacted, we do not discuss "non-reacting" ions. In fact, FT-ICR-MS time is expensive and analyses are time consuming. Thus we focused our instrument time on the reactive compounds. Since we have not chemically characterized these species, we cannot hypothesize why these ions are not oxidized by OH. The organic species with "reactant-like" trends account for about 30% of the total ion abundance observed in the mist chambers. This is now indicated in the text, line 346:

**"Together, the ions discussed herein account for 30% of the total ion abundance."**

The organic acids observed in this study are formed as intermediates or products and as the reviewer says, OH oxidation will lead to their formation. It is certainly possible that other acids could be formed.

We agree with the reviewer that it would be helpful to provide an upper limit for the amount of aqueous SOA that can be formed from cloud/fog processing in a given sample. In doing so, we would have to make quite a few assumptions such as collection of all gasphase WSOC, conversion of all SOA to oxalate/pyruvate etc. that could greatly distort the results and mislead the community, and hence we have opted to not provide such an estimate. A technique that could provide a better estimate would be to conduct droplet evaporation experiments on the collected samples.

Specific comments

The authors state in the methods section that intensive days were chosen because of high concentration predictions by NCAR and that during those days, they observed higher than usual TOC content in general. However, in examining Supplementary Table S1, it seems that the days marked "intensive" are not much higher than others, if at all. The range given in the text (92-179  $\mu$ M-C) does not match the range of TOC on intensive days. Can this be clarified? Does this range only represent the days that were used in further measurements?

We apologize for this confusion. To clarify, we primarily focused on the intensive days that had the highest TOC contents; primarily the first three intensive periods highlighted in the table. Besides these days, we also chose to pick July 20th and 21st to perform experiments due to the high TOC values on these days. We have clarified this in the text, line 152:

"In general, mist chamber samples on intensive sampling days had higher organic content (TOC = 92-179  $\mu$ M-C), and hence we focused our experiments on those days and included two additional days from the non-intensive period that had high TOC values (Table 1)."

**Line 208: Why is glyoxal the only standard compared across all analysis days? Were the variabilities of the other standards on a similar scale? If so, this should be stated here.**

Similar to glyoxal, the other standards also showed a similar behavior. Glyoxal variance as mentioned in the text is

**should be explained the first time it is seen.**

We thank the reviewer for this suggestion and have included this information in the text for m/z 187 as follows:

"The MS-MS shows loss of methanol, to give a product ion at m/z 155.0680 corresponding to the neutral molecular formula,  $C_6H_{12}O_{3,}$  consistent with expectations for a sodium ion complex of a dihydroxy hemiacetal in methanol solution, shown in Figure 2."

In line 319 and again in lines 360-364, the authors state that they expect IEPOX, ISOPOOH, and glyoxal were lost in their experiments during sampling or storage. Have any tests been done to quantify losses of other compounds during sampling or storage? If these compounds were lost, it seems likely that there are others that are either lost or not fully recovered. Were any recovery studies done with known amounts of standards instead of spiking the samples just before analysis?

We have not performed recovery studies, although this would be a good addition to our future work. However, we have now added a section called "Methodological limitations" of the work that focuses on issues such as losses, line 318:

"However, WSOGs can be lost during sampling and storage through: 1) losses in tubing and by adsorption to the QFF during collection, 2) reactions in the mist chamber during collection with water-soluble ambient oxidants capable of penetrating the inlet (e.g., ozone), and 3) losses during storage post collection. The QFF removes particles upstream of the mist chambers. In the early stages of sampling, on the clean filter, adsorption of gases on the filter will reduce the concentrations of gases sampled by the mist chamber until these gases reach gas phase – adsorbed phase equilibrium. Using glyoxal as a WSOG-surrogate and the work of (Mader and Pankow, 2001) we predict that the measured WSOG in the mist chamber will be depleted for less than 2% of our sampling time (after  $

Membrane Filters and Quartz Fiber Filters, Environmental Science & Technology, 35, 3422-3432, 10.1021/es0015951, 2001.

Martinelango, P. K., Dasgupta, P. K., and Al-Horr, R. S.: Atmospheric production of oxalic acid/oxalate and nitric acid/nitrate in the Tampa Bay airshed: Parallel pathways, Atmospheric Environment, 41, 4258-4269,

http://dx.doi.org/10.1016/j.atmosenv.2006.05.085, 2007.

Nguyen, T. B., Crounse, J. D., Teng, A. P., St. Clair, J. M., Paulot, F., Wolfe, G. M., and Wennberg, P. O.: Rapid deposition of oxidized biogenic compounds to a temperate forest, Proceedings of the National Academy of Sciences, 112, E392-E401, 10.1073/pnas.1418702112, 2015.

Paulot, F., Crounse, J. D., Kjaergaard, H. G., Kurten, A., St Clair, J. M., Seinfeld, J.H., and Wennberg, P. O.: Unexpected Epoxide Formation in the Gas-PhasePhotooxidation of Isoprene, Science, 325, 730-733, DOI 10.1126/science.1172910, 2009.

Tan, Y., Perri, M. J., Seitzinger, S. P., and Turpin, B. J.: Effects of Precursor Concentration and Acidic Sulfate in Aqueous Glyoxal–OH Radical Oxidation and Implications for Secondary Organic Aerosol, Environmental Science & Technology, 43, 8105-8112, 10.1021/es901742f, 2009.

Yao, X., Fang, M., and Chan, C. K.: Size distributions and formation of dicarboxylic acids in atmospheric particles, Atmospheric Environment, 36, 2099-2107, http://dx.doi.org/10.1016/S1352-2310(02)00230-3, 2002.

Zhang, Z., Lin, Y. H., Zhang, H., Surratt, J. D., Ball, L. M., and Gold, A.: Technical Note: Synthesis of isoprene atmospheric oxidation products: isomeric epoxydiols and the rearrangement products cis- and trans-3-methyl-3,4dihydroxytetrahydrofuran, Atmos. Chem. Phys., 12, 8529-8535, 10.5194/acp-12-8529-2012, 2012.

Zhang, H., Zhang, Z., Cui, T., Lin, Y.-H., Bhathela, N. A., Ortega, J., Worton, D. R., Goldstein, A. H., Guenther, A., Jimenez, J. L., Gold, A., and Surratt, J. D.: Secondary Organic Aerosol Formation via 2-Methyl-3-buten-2-ol Photooxidation: Evidence of Acid-Catalyzed Reactive Uptake of Epoxides, Environmental Science & Technology Letters, 1, 242-247, 10.1021/ez500055f, 2014.

**Identifying precursors and aqueous organic aerosol formation pathways during the SOAS campaign**

5

Neha Sareen1, Annmarie G. Carlton1, a, Jason D. Surratt2, Avram Gold2, Ben Lee3, Felipe

D. Lopez-Hilfiker3, b Claudia Mohr3, c Joel A. Thornton3, Zhenfa Zhang2, Yong B. Lim1, d Barbara J. Turpin2

 Department of Environmental Sciences, Rutgers University, 14 College Farm Road, New Brunswick, New Jersey 08901, United States

 2. Department of Environmental Sciences and Engineering, Gillings School of Public Health, University of North Carolina at Chapel Hill, Chapel Hill, North Carolina 27599, United States

3. Department of Atmospheric Sciences, University of Washington, Seattle, Washington 98195 United States

a Now at Department of Chemistry, University of California, Irvine, CA 92697, USA

15 Now at Laboratory of Atmospheric Chemistry, Paul Scherrer Institute, 5232 Villigen PSI, Switzerland Now at Institute of Meteorology and Climate Research, Atmospheric Aerosol Research, Karlsruhe Institute of Technology, Karlsruhe, Germany

 $\frac{d}{v}$ Now at Center for Environment, Health and Welfare Research, Korea Institute of Science and Technology, Seoul 02792, Republic of Korea,

20 Correspondence to: Neha Sareen (neha.sareen15@gmail.com), Barbara J. Turpin (bjturpin@email.unc.edu)

Abstract. Aqueous multiphase chemistry in the atmosphere can lead to rapid transformation of organic compounds, forming highly oxidized low-volatility organic aerosol and, in some cases, light-absorbing (brown) carbon. Because liquid water is globally abundant, this chemistry could substantially impact climate, air quality, and

25 health. Gas-phase precursors released from biogenic and anthropogenic sources are oxidized and fragmented, forming water-soluble gases that can undergo reactions in the aqueous phase (in clouds, fogs, and wet aerosols) leading to the formation of secondary organic aerosol (SOAAQ). Recent studies have highlighted the role of certain precursors

| Neha Sareen 8/27/2016 3:14 PM |
|-------------------------------|
| Formatted: Superscript        |
| Neha Sareen 8/27/2016 3:16 PM |
| Deleted: a         |
| Neha Sareen 8/27/2016 3:16 PM |
| Deleted: b         |
| Neha Sareen 8/27/2016 3:16 PM |
| Deleted: °                    |
| Neha Sareen 8/28/2016 4:19 PM |
| Deleted: *                    |

[revised manuscript text omitted]
 Neha Sareen 8/27/2016 3:16 PM Deleted: longer Neha Sareen 8/27/2016 3:16 PM Deleted: All the peaks

**Neha Sareen 8/27/2016 3:16 PM**

Deleted: As discussed above, all precursor ions appear in the positive ion mode of the ESI-MS, consistent with the presence of carbonyl compounds and polyols . They are odd ions, suggesting they are likely not nitrogen-containing species. Neha Sareen 8/27/2016 3:16 PM Deleted: (using Neha Sareen 8/27/2016 3:16 PM Deleted: ) and Neha Sareen 8/27/2016 3:16 PM Deleted: corresponding Neha Sareen 8/27/2016 3:16 PM Deleted: from Neha Sareen 8/27/2016 3:16 PM Deleted: No fragmentation occurred Neha Sareen 8/27/2016 3:16 PM

**Deleted:** are shown in Figures 2 and 3. Possible structures for the other ions are shown in Figure 5. As seen in the proposed structures, the parent compounds contain multiple –OH groups, suggesting that they are polyols or aldehydes in the gas-phase and are hydrated with water or methanol in our ESI-MS.

**Neha Sareen 8/27/2016 3:16 PM**
* * *
Neha Sareen 8/27/2016 3:16 PM

**Deleted:** Na+ or H+.) The daughter ion peak seen during fragmentation of the positive ion at m/z 187 corresponds to m/z 155.0680, CH3OH loss, corresponding to the molecular formula,  $C_6H_{12}O_3$  We propose that this compound is present in the gas-phase as the C6H12O3 aldehyde and in water as the C6H14O4 tetrol shown in Figure 2. C6H12O3 is consistent with an oxidation product of E-2-hexenal and Z-3hexenal, both being unsaturated aldehydes that have frequently been detected during field studies and are emitted to the atmosphere from vegetation due to leaf wounding (O'Connor et al., 2006). The oxidation mechanism of these two green leaf volatiles to form C6H12O3 is shown in Figure 3.

 $C_7H_{16}O_4Na$  is assigned by the Midas molecular formula calculator based on the exact mass 187.0942 from FT-ICR MS, corresponding to the composition  $C_7H_{16}O_4$  for the 390 neutral compound. The MS-MS shows loss of methanol, to give a product ion at m/z155.0680 corresponding to the neutral molecular formula, C6H12O3, consistent with expectations for a sodium ion complex of a dihydroxy hemiacetal in methanol solution, shown in Figure 2. In the absence of methanol, this compound would appear hydrated with water as the  $C_6H_{14}O_4$  tetrol, as shown in the blue box in Figure 2. The corresponding 395 gas-phase compound is shown in the tan box in Figure 2. E-2-hexenal and Z-3-hexenal are unsaturated aldehydes that have frequently been detected during field studies and are emitted to the atmosphere from vegetation due to leaf wounding (O'Connor et al., 2006). The gas-phase oxidation of these two green leaf volatiles, as shown in Figure 4a and 4b, could explain the presence of C6H12O3 in the gas phase and C6H14O4 in the aqueous phase 400 (Figure 2).

m/z 173: On most sampling days this reactant mass has the highest abundance in the positive mode ESI mass spectra (Supplementary Figure S1). Similar to other reactant peaks, it reacts away within the first 40 minutes of exposure to OH in the cuvette chamber (Figure 1). The Midas-suggested molecular formula for this parent ion (m/z173.0782) and its two fragment ions at m/z 141.0523 and 129.0524 are C6H14O4, C5H10O3, and C4H10O3, respectively (a reactive parent ion with the formula C4H10O3 was also observed, and is discussed below).

A likely structure for positive mode m/z 173 is shown in Figure 3. In this case the compound is proposed to be a  $C_5H_{10}O_3$  aldehyde in the gas phase (tan box in Figure 3) and a  $C_5H_{12}O_4$  tetrol in water (blue box in Figure 3). In the FT-ICR-MS it is seen

| Neha Sareen 8/27/2016 3:16 PM       |
|-------------------------------------|
| Deleted: ion                        |
| Neha Sareen 8/27/2016 3:16 PM       |
| Deleted: ESI-MS operated in         |
| Neha Sareen 8/27/2016 3:16 PM       |
| Deleted: ion                        |
| Neha Sareen 8/27/2016 3:16 PM       |
| Deleted: the                        |
| Neha Sareen 8/27/2016 3:16 PM       |
| Deleted: mass                       |
| Ne 8/27/2016 4:29 PM                |
| Deleted: A summer durith surglamber |

Deleted: A compound with a molecular formula of C5H10O3 (the same mass as a fragment ion discussed above) was observed in the gas-phase in the same location (Centerville field site) by high-resolution time-of-flight chemical ionization mass spectrometry (HRToF-CIMS) (Lee et al., 2014), coupled to a filter inlet for gases and aerosols (FIGAERO) (Lopez-Hilfiker et al., 2015; Lopez-Hilfiker et al., 2014). This could be the same compound that we measure as m/z 173.0782. as explained below. Note that the HRToF-CIMS employed iodide ionization, which forms organic-iodide adducts, resulting in a virtually fragmentation free ionization. Gas phase measurements from the HRToF-CIMS were made in real time through a 3/4" PTFE inlet operated at 16 standard L min-1. Isoprene hydroxy hydroperoxide (ISOPOOH) and isoprene epoxide (IEPOX) are both detected at this mass, but HRToF-CIMS is more sensitive to ISOPOOH. ISOPOOH is an OH oxidation product of isoprene, which is further oxidized by OH under low-NO conditions to form isomeric isoprene epoxydiols (IEPOX) (Paulot et al., 2009). Both IEPOX and ISOPOOH are prevalent at the SOAS ground site due to the abundance of isoprene emissions in this forested region but it is likely that m/z 173 is not indicative of these two compounds in our samples. As discussed below in detail, based on ESI-MS measurements with an IEPOX standard, it can be confirmed that IEPOX was not detected in the mist chamber samples. The O-O peroxide bond in ISOPOOH is the weakest bond in the molecule, and hence when undergoing MS-MS, should be the first to fragment. There is no evidence of this bond breaking in the fragmentation spectra for m/z173, leading to the conclusion that the detected compound is not ISOPOOH. IEPOX and ISOPOOH were present in the ambient air at the field site. We expect that they were lost during sampling or storage.

**Neha Sareen 8/27/2016 3:16 PM**

Neha Sareen 8/27/2016 3:16 PM Deleted:

405

410

hydrated with methanol. The parent ion at m/z 173 loses methanol to form C5H10O3 (m/z 141.0523), and it also loses C2H4O to form C4H10O3 (m/z 129.0524). The aqueous oxidation precursor observed as m/z+ 173 and tentatively identified as the C5H10O3

aldehyde shown in tan in Figure 3, could be derived from another green leaf volatile.
Specifically, C5H10O3 is consistent with the gas-phase oxidation product of (*E*)-2-methyl2-butenal, another green leaf volatile (Figure 4c) (Jiménez et al., 2009; Lanza et al.,
2008), It has also been reported as an isoprene oxidation product (Yu et al., 1995).

Positive ions at m/z 143, 129, and 125: No fragments were observed for these
reactants under conditions of MS-MS acquisition in this work. The Midas-predicted molecular formulae for the ions at m/z 143.0676, 129.0520, and 125.096 are C5H12O3, C4H10O3, and C8H12O, respectively. Interestingly, the reactant detected at m/z 129 has the same mass as a fragment of the parent ion at m/z 173 discussed earlier (and the structure of the C4H10O3 fragment shown in Figure 3 is a possible structure for m/z 129).

475Methodological Limitations: In this work, we aim to collect the ambient mix of
water-soluble gases into water at concentrations comparable to those found in clouds and
fogs with the purpose of simulating cloud/fog-relevant OH oxidation chemistry and
identifying previously unrecognized precursors of aqueous chemistry. Below we discuss
limitations with respect to our ability to collect and store these aqueous mixtures and with

480 respect to our ability to identify the OH-reactive compounds collected.

Mist chamber collection times (4 hr) were selected with the aim of collecting ambient mixtures of water-soluble gases near Henry's law equilibrium. Two pieces of evidence suggest that gas-aqueous partitioning of the water-soluble organic gases is close to Henry's Law equilibrium in our samples. In previous testing conducted in a different

**Neha Sareen 8/27/2016 3:16 PM Deleted: 3c Neha Sareen 8/27/2016 3:16 PM Deleted: ; Lanza et al., 2008)**

Neha Sareen 8/27/2016 3:16 PM Deleted:

Neha Sareen 8/27/2016 3:16 PM Formatted: Font:Not Italic Neha Sareen 8/27/2016 3:16 PM

**Neha Sareen 8/27/2016 3:16 PM**

[revised manuscript text omitted]
equivalents |
|--------------------------------------|---------------------------------------------------|-----------------------------------------------------------------------|----------|----------------------------|
| 187                                  | 187.0942                                          | C7H16O4Na                                                             | 164.1043 | 0                          |
| 107                                  | 155.0680                                          | C 6 H 12 O 3 Na                      | 132.0786 | 1                          |
|                                      | 173.0782                                          | C 6 H 14 O 4 Na                      | 150.0887 | 0                          |
| 173                                  | 141.0523                                          | C 5 H 10 O 3 Na                      | 118.0625 | 1                          |
|                                      | 129.0524                                          | C 4 H 10 O 3 Na                      | 106.0625 | 0                          |
| 143                                  | 143.0676                                          | C 5 H 12 O 3 Na                      | 120.0781 | 0                          |
| 129                                  | 129.0520                                          | C 4 H 10 O 3 Na                      | 106.0625 | 0                          |
| 125                                  | 125.096                                           | C 8 H 13 O                                      | 124.0883 | 3                          |

Neha Sareen 8/27/2016 3:16 PM Deleted: equivalence

1025

---

## Author Response (AR2)

Response to comments

*The authors have largely addressed my main comments, with additional discussion of uncertainties and limitations. Additional comments (related to the revised text) are below; once these are addressed, this paper should be published in ACP.*

*Line 246: It's unclear what these "field water" blanks are – is it just clean water treated as sample? In that case, how can the authors be sure there are no organic compounds introduced from the sampling line, filter, etc? These could be potentially major sources of organic contaminants.*

Field blanks are water collected from the same source at the same time as the water used in the mist chambers. Experiments conducted with field water blanks ensure that reactive water-soluble gases that we highlight in this work are not contaminants from the water itself. It was not possible to sample zero air through the Teflon inlet tubing and mist chambers, as the reviewer initially suggested because of the high flow rates needed (100 lpm). The mist chambers were baked before use and Teflon tubing is the standard for use as an inlet material for measurement of polar organic gases, including for CIMS. While some compounds are known to partition to Teflon (Krechmer et al., 2016; reference included in manuscript), the structures we highlight in this work are not likely derived from Teflon. Although we are confident that these structures are not contaminants, we agree that it is important to be precise about what we know definitively, which is that they are not contaminants from the water source. Thus, we have added the following:

Line 140 currently reads: "The mist chambers were operated with 25 mL of $17.5 \pm 0.5$ M$\Omega$ ultra-pure water; additional water was added during the run to replace water lost by evaporation. Samples from all four mist chambers were composited daily and frozen in $35 - 40$ mL (experiment-sized) aliquots."

We have added: "Field water blanks were collected concurrently from the same water supply, transported and stored with samples."

On line 145 it reads: "Prior to and at the end of a sampling day, each mist chamber was cleaned"

Before this sentence, we added "at the beginning of the study, mist chambers were baked at $500^{o}$C for 4 hours"

On line 247 it currently reads: "In control experiments where we generated OH radicals in field water blanks, these ions were not observed, confirming that they are not contaminants."

We have changed this to read "In control experiments where we generated OH radicals in field water blanks, these ions were not observed, confirming that they are not contaminants from the water source."

*Line 274: The diol-formation mechanism put forth by Zhang 2014 for MBO is highly speculative and has not been established as an important mechanism in alkene oxidation. It was put forth as a way to explain the formation of certain SOA tracer species, but no mechanistic evidence has ever been presented. (In the scheme given, the 1,5-shift shown is quite endothermic so is unlikely to be important.) The uncertainties related to diol formation mechanisms should at least be mentioned.*

The diols are proposed to result from hydrolysis of the corresponding epoxides, which may be generated by attack of OH followed by addition of $O_2$, 1,5-H transfer and elimination of OH[1,2] or by reaction with $RO_2$.[3]

The references below have been added to the manuscript. And we have modified the figure caption to explain that this is proposed, rather than established, and why we think it makes sense:

"**Figure 4**: Gas-phase oxidation of (a) *E*-2-hexenal and (b) *Z*-3-hexenal and (c) (*E*)-2-methyl-2-butenal. The diols are proposed to result from hydrolysis of the corresponding epoxides, which may be generated by attack of OH followed by addition of $O_2$, 1,5-H transfer and elimination of OH[1,2] or by reaction with $RO_2$.[3]"
* * *
1. John D. Crounse, Hasse C. Knap, Kristian B. Ørnsø, Solvejg Jørgensen, Fabien Paulot, Henrik G. Kjaergaard, and Paul O. Wennberg, "Atmospheric Fate of Methacrolein. 1. Peroxy Radical Isomerization Following Addition of OH and O2", *J. Phys. Chem. A* **2012**, *116*, 5756−5762.

**Scheme 3. Detail of 1,4- and 1,5-H-Shift Reactions of MACR-1-OH-2-OO and the Ensuing Decompositions**

2. Jozef Peeters, Luc Vereecken, Gaia Fantechi, "The detailed mechanism of the OH-initiated atmospheric oxidation of a-pinene: a theoretical study" *Phys. Chem. Chem. Phys.*, **2001**, *3*, 5489±5504

3. Moray S. Stark, "Epoxidation of Alkenes by Peroxyl Radicals in the Gas Phase: Structure-Activity Relationships", *J. Phys. Chem. A* **1997,** *101,* 8296-8301.

$$\text{ROO}^\bullet + \overset{\diagdown}{\diagup}C = C\overset{\diagup}{\diagdown} \underset{k_{-1}}{\overset{k_1}{\rightleftharpoons}} \text{R}-\text{O}-\text{O}-\overset{\diagdown}{\diagup}C-C\overset{\diagup}{\diagdown}{}^\bullet \overset{k_2}{\longrightarrow} \text{RO}^\bullet + \overset{\diagdown}{\diagup}C\overset{O}{\diagdown}\text{C}\overset{\diagup}{\diagdown}$$

*Line 347: The authors now note a full 70% of the ion signal did not decrease with oxidation. I agree that some changes may be masked by the simultaneous formation and loss of ions a given molecular formula ion, the odds of these exactly cancelling for ALL ions is extremely small. So these almost certainly includes unreactive species, which needs to be commented upon in more detail – perhaps an upper limit for the OH rate constant (given the estimated OH levels) for comparison. The fact that there appears to be a substantial fraction of dissolved organic species which do not rapidly oxidized is a very significant (and surprising) result in the context of atmospheric aqueous chemistry, and so simply should not be ignored.*

We have found that when we conduct experiments where we oxidize mixtures it can be difficult to definitively identify compounds as reactive that we know to be because they are being simultaneously formed and oxidized, yielding a series of modest increases and decreases in signal strength. When we model the chemical dynamics that we expect to see for these mixtures, see similar concentration dynamics. So we do think this is a possibility.

We have also made it more clear that this 70% also may include compounds that react only slowly in water under our experimental conditions.

Line 346 currently says: "Together, the ions discussed herein account for 30% of the total ion abundance. The remaining 70% did not exhibit a clearly decreasing trend during OH oxidation experiments. It should be recognized that some water-soluble OH-reactive compounds might not have decreased during experiments because they were intermediates of multiple precursors."

This now reads: "Together, the ions discussed herein account for 30% of the total ion abundance. The remaining 70% did not exhibit a clearly decreasing trend during OH oxidation experiments. The remaining 70% could represent compounds that are stable with respect to OH oxidation in water or that react too slowly to observe under the experimental conditions used. However, it should be recognized that some water-soluble OH-reactive compounds might not have been identifiable as reactive because they were intermediates of multiple precursors. We have observed this phenomenon previously in aqueous OH oxidation experiments conducted with mixed aldehyde standards (unpublished). In those experiments, several modest concentration increases and decreases were observed in experiments and model predictions for some intermediates."

*SI: Here it is noted that CIMS sensitivity is highly sensitive to structure - however so is ESI! That is probably worth mentioning here; this makes the comparison between the two techniques potentially even more challenging.*

We have added the following statement to the SI:
"Additionally, both techniques are highly sensitive to the structure of the compound, making comparisons challenging."

*References:*

[revised manuscript text omitted]

$m/z+$ 173

$-CH_3OH$
$-32$

$C_6H_{14}O_4$

$Na^+$

$-C_2H_4O$
$-44$

$C_5H_{10}O_3$

$Na^+$

$C_4H_{10}O_3$

$Na^+$

$C_5H_{10}O_3$ gas-phase

$C_5H_{12}O_4$ aqueous-phase

810 **Figure 3**. Proposed structure for the positive ion at *m/z* 173. The top structures in each panel are the parent compound detected as a reactant in the ESI-MS; the following structures show the MS/MS fragments. This compound would take the forms shown in the shaded boxes when present in atmospheric air and water.

815   **(a)**

**(b)**

**(c)**

820

**Figure 4**: Gas-phase oxidation of (a) *E*-2-hexenal and (b) *Z*-3-hexenal and (c) (*E*)-2-methyl-2-butenal. The diols are proposed to result from hydrolysis of the corresponding epoxides, which may be generated by attack of OH followed by addition of $O_2$, 1,5-H transfer and elimination of OH (Crounse et al., 2012; Peeters et al., 2001) or by reaction

825   with $RO_2$. (Stark, 1997)

[Figure]

**Figure 5**. (a) Oxalate (by IC) for all OH radical oxidation experiments conducted with ambient samples (Table 1). (b) Abundance of the negative ion at *m/z* 87 (pyruvate) as observed in the ESI-MS when the ambient SOAS samples are exposed to OH. Error bars represent the pooled coefficient of variation calculated across experimental days. Note that oxalate and pyruvate are formed in all samples in the presence, but not the absence, of OH. Gray points represent control experiments (June 11 sample + UV, June 11 sample + $H_2O_2$, June 30 field water blank + OH).